# Genetically stable CRISPR-based kill switches for engineered microbes

Austin G. Rottinghaus[1,7], Aura Ferreiro [2,3,7], Skye R. S. Fishbein[2,4], Gautam Dantas [2,3,4,5,6✉] & Tae Seok Moon [1,6✉]

Microbial biocontainment is an essential goal for engineering safe, next-generation living therapeutics. However, the genetic stability of biocontainment circuits, including kill switches, is a challenge that must be addressed. Kill switches are among the most difficult circuits to maintain due to the strong selection pressure they impart, leading to high potential for evolution of escape mutant populations. Here we engineer two CRISPR-based kill switches in the probiotic *Escherichia coli* Nissle 1917, a single-input chemical-responsive switch and a 2-input chemical- and temperature-responsive switch. We employ parallel strategies to address kill switch stability, including functional redundancy within the circuit, modulation of the SOS response, antibiotic-independent plasmid maintenance, and provision of intra-niche competition by a closely related strain. We demonstrate that strains harboring either kill switch can be selectively and efficiently killed inside the murine gut, while strains harboring the 2-input switch are additionally killed upon excretion. Leveraging redundant strategies, we demonstrate robust biocontainment of our kill switch strains and provide a template for future kill switch development.

[1] Department of Energy, Environmental and Chemical Engineering, Washington University in St. Louis, St. Louis, MO, USA. [2] The Edison Family Center for Genome Sciences & Systems Biology, Washington University School of Medicine, St. Louis, MO, USA. [3] Department of Biomedical Engineering, Washington University in St. Louis, St. Louis, MO, USA. [4] Department of Pathology and Immunology, Washington University School of Medicine, St. Louis, MO, USA. [5] Department of Molecular Microbiology, Washington University School of Medicine, St. Louis, MO, USA. [6] Division of Biology and Biomedical Sciences, Washington University in St. Louis, St. Louis, MO, USA. [7]These authors contributed equally: Austin G. Rottinghaus, Aura Ferreiro. ✉email: dantas@wustl.edu; tsmoon@wustl.edu

Probiotic microbes have become effective chasses for engineering diagnostic and therapeutic technologies. One of the most commonly engineered probiotic strains is *Escherichia coli* Nissle 1917 (EcN). Engineered strains of EcN have been successfully used to diagnose and treat bacterial infections[1,2], cancers[3–5], gastrointestinal bleeding[6], inflammatory disorders[7–9], and obesity[10] in a variety of animal models. EcN strains engineered to treat metabolic disorders are being evaluated in human clinical trials with promising early-phase results[11,12]. However, there are important safety concerns associated with organisms genetically engineered for medical applications. Probiotics are living organisms that have the potential to mutate and evolve undesirable traits over the course of diagnosis or treatment. Such adaptations can include loss of beneficial functions of the engineered system, gain of deleterious functions such as competitive exclusion of native microbes, pathogenic potential against the host, or environmental contamination if they spread outside the host[13–16]. To mitigate these concerns, engineered probiotics should possess biocontainment systems that enable both selective removal from the host and prevent their environmental dissemination[17].

Biocontainment circuit designs are focused on preventing proliferation in the wild, and typically involve an input that is specific to the permissive environment and repressive to the killing circuit, such that upon exit of the permissive environment, the lethal components are expressed[18]. Several such biocontainment strategies have been developed with varying degrees of efficacy and stability, including use of auxotrophy[11,19,20] and synthetic amino acids[21–23]. While approaches like synthetic auxotrophy are evolutionarily stable in that they do not readily give rise to escape mutants[21], a limitation of these methods is that they may require the probiotic environment to be supplemented with additional survival factors ('permissive molecules'). Completely withholding these molecules in the gut, for example by administering an auxotrophic strain without the essential compound, effectively limits probiotic lifespan in vivo[11], but it may also limit therapeutic potential depending on the rate of probiotic cell death in the absence of the permissive molecules. Alternatively, the permissive molecules may theoretically be supplied to patients in conjunction with the probiotic, but this design complicates administration as well as the selective removal of probiotics from the gut since the time to full clearance of the permissive molecules may be difficult to control. In addition, if the permissive molecules are natively present in the gut, these circuits can be compromised by cross-feeding[20,21,24].

An inverse kill switch design would then be one in which the baseline state in the gut is permissive without supplementation of exogenous molecules; correspondingly, the lethal components of the circuit are expressed in response to supplied inducers, or environmental signals external to the gut. Numerous genetic circuits have been developed that initiate cell death in response to a chemical inducer[25–28]. Similarly, biocontainment circuits have been developed in *E. coli* using temperature sensors tuned to differentiate physiological and environmental temperatures[18,29,30]. These kill switches control cell survival using a variety of mechanisms, including expression of toxins and lysis proteins[18,25–27], degradation of essential proteins[26], and cleavage and degradation of the genome by Cas3 proteins[28]. Both temperature-sensitive circuits designed by Piraner et al. and Stirling et al. used the *E. coli* CcdB-CcdA toxin-antitoxin system to control cell survival. The kill switch engineered by Piraner et al. used a modified version of the *Salmonella typhimurium*-native $P_{tlpA}$-*tlpA* sensor and achieved a 4-log reduction in fecal cell number[30], while the kill switch engineered by Stirling et al. used the *E. coli*-native $P_{cspA}$ promoter and achieved a 5-log reduction in fecal cell number[29]. Notably, functional redundancy offered by the combination of the $P_{cspA}$-controlled temperature-sensitive kill

switch with an orthogonal pH-sensitive kill switch mechanism synergistically improved in vitro killing efficacy such that surviving colony counts were below the 11-log limit of detection[18]. However, kill switches that induce cell death by expressing toxins, lysis proteins, and proteases are prone to mutational inactivation, often leading to population dominance of non-functional variants, or have not been characterized for genetic stability[26]. To overcome this stringent evolutionary selection, such kill switch systems must be designed to be highly stable. The temperature-inducible toxin-antitoxin kill switch engineered by Stirling et al. was shown to be stable over 140 generations of growth in vitro and at least 10 days of growth in the mouse gut[29], while the combined temperature- and pH-inducible kill switch was stable over at least 100 generations in vitro[18]. In contrast, a CRISPR-Cas3-based system has been shown to be stable for 1700 generations when applied to plasmid removal[28]. However, it is unclear whether the same stability would persist if the system was applied to cell death.

To engineer a genetically stable probiotic with viability that is controllable both inside and outside the host, we used a step-wise design approach to develop two CRISPR-Cas9-based kill switches (CRISPRks) in EcN. First, we built a single-input kill switch that can initiate probiotic death in response to the chemical inducer anhydrotetracycline (aTc). After iterative optimization for stability and efficacy, this circuit became the foundation for a 2-input kill switch that can additionally initiate death in response to the temperature decrease that occurs upon excretion from the host. Both designs allow the engineered microbe to be selectively removed in situ from the gut, while the final 2-input CRISPRks additionally prevents the microbe from surviving outside the body. To achieve genetic stability of the kill switches, we applied parallel approaches of genetic engineering and environmental control, including functionally redundant expression cassettes, antibiotic-free plasmid maintenance systems, knockouts of key drivers of DNA-mutagenesis in the SOS response, and provision of intra-niche microbial competition. Both kill switches exhibited significant long-term stability, with efficient killing maintained after 28 days (224 generations) of continuous growth in vitro. In mice, we innovatively leveraged intra-niche competitive inhibition to selectively disadvantage the target EcN subpopulation[31]. The single-input CRISPRks allowed the engineered probiotic to be completely eliminated from the mouse gut in response to aTc consumption. Similarly, the 2-input CRISPRks achieved virtually complete eradication of the probiotic population when both chemical- and temperature-induction were applied. The principles and genetic parts used in the CRISPRks described here (i.e., Cas9 and guide RNA) have been employed in various microbial species[32–34], in contrast to the limited use of Cas3 in microbes, highlighting the potential for deploying this generalizable biocontainment platform to other engineered probiotic and microbial strains for diverse applications.

## Results
**Development of a chemical-inducible CRISPR-based kill switch for EcN.** We first designed a kill switch that induces EcN cell death in response to the chemical inducer aTc (Fig. 1a; Table 1). To initiate cell death, we utilized a CRISPR/Cas9-based approach. *E. coli* can repair double-stranded DNA breaks (DSBs), as caused by Cas9, through RecA-dependent homologous recombination with sister genomes generated by DNA replication[35]. However, the damage caused by Cas9 is lethal if each copy of the replicating genome is cut[36]. To make CRISPR-based cell death dependent on the presence of aTc, we expressed Cas9 from a low-copy plasmid and genome-targeting guide RNAs (gRNAs) from a medium-copy plasmid using aTc-inducible $P_{tet}$ promoters.

We next sought a gRNA that could target the genome and initiate cell death with high efficiency. We hypothesized that

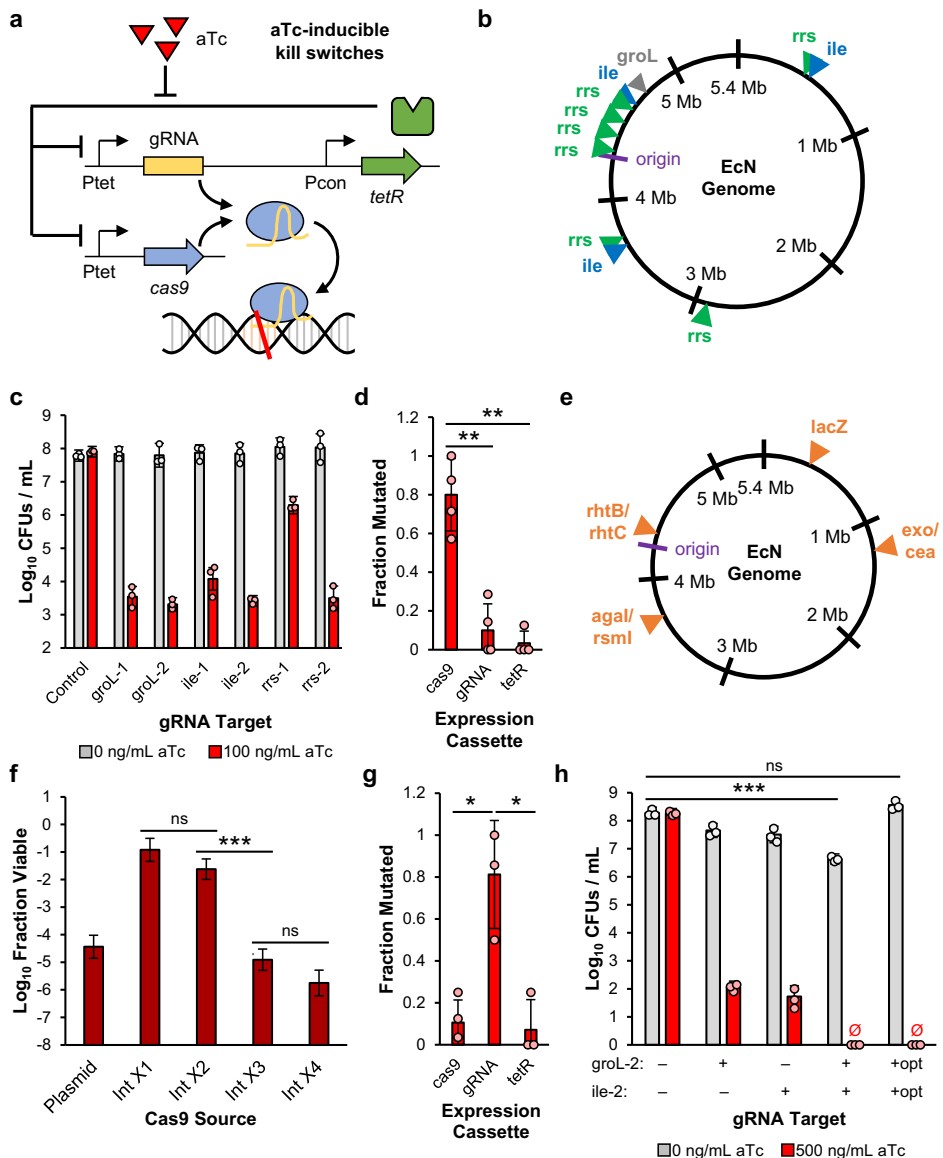

**Fig. 1 Development of a chemical-inducible CRISPR-based kill switch for EcN. a** Schematic of the aTc-inducible CRISPR-based kill switch. Cas9 and gRNAs are expressed on independent plasmids by aTc-inducible $P_{tet}$ promoters. TetR, which regulates the expression of $P_{tet}$ promoters in an aTc-dependent manner, is constitutively expressed on the gRNA plasmid. In the presence of aTc, TetR is unable to bind to its target promoters, leading to expression of Cas9 and the gRNAs. The Cas9/gRNA complex then binds to and cleaves its genomic targets, leading to cell death. **b** gRNA target locations in the EcN genome: the single copy *groL* gene (gray), the three copy *ileTUV* genes (blue), and the seven copy *rrsABCDEGH* genes (green). **c** $Log_{10}$ CFUs for the no gRNA control and six gRNA expression plasmids in wild-type EcN with the $P_{tet}$-*cas9* expression plasmid. **d** Sequencing results from 30 survivors from **c** with non-functional ile-2 kill switches. **e** Genomic neutral integration sites used for $P_{tet}$-*cas9* integrations: within the *lacZ* gene, between *rhtB* and *rhtC*, between *agaI* and *rsmI*, and between *exo* and *cea*. **f** $Log_{10}$ Fraction Viable for EcN strains with plasmid-based, one genome-integrated (Int X1), two genome-integrated (Int X2), three genome-integrated (Int X3), and four genome-integrated (Int X4) $P_{tet}$-*cas9* expression cassettes. All strains contain the same ile-2 gRNA plasmid. **g** Sequencing results from 24 survivors from **f** with non-functional Int X4 kill switches. **h** $Log_{10}$ CFUs for EcN with four genomic $P_{tet}$-*cas9* integrations and different combinations of the groL-2 and ile-2 gRNAs. '+opt' represents the respective $P_{tet}$-gRNA cassette with an optimized $P_{tet}$ promoter. aTc concentration was increased to 500 ng/mL, from 100 ng/mL in **c**, based on the transfer curve characterized in Supplementary Fig. 1e. For all sequencing, the fraction mutated is the fraction of total sequenced cassettes that contained a mutation. For all kill switch assays, exponential phase cells for each strain were induced with 0 and 100 or 500 ng/mL aTc for 1.5 h. Values and error bars are the average and standard deviation of biological triplicate, respectively. Statistical comparisons were performed using two-tailed unpaired *t*-tests (*$P < 0.05$; **$P < 0.01$; ***$P < 0.001$). See also Supplementary Fig. 1. Source data with p-values are provided as a Source Data file.

gRNAs with multiple genomic binding sites would have higher killing efficiencies as the gRNA-Cas9 complex would have a higher probability of locating a target site. In addition, multi-locus genome cleavage should decrease the probability of DSB rescue through homologous recombination[37], or genomic mutations in the target sites rendering the cell immune to the kill

switch. To explore the effect of target copy number, we tested the aTc-response of six total gRNAs for three target genes present at various copies throughout the genome: the single-copy *groL* gene, the three-copy *ileTUV* genes, and the seven-copy *rrsABCDEGH* genes (Fig. 1b, c). To quantify the efficiency for each gRNA, we defined the term 'fraction viable' as the ratio of colony forming

**Table 1 Names and descriptions of each no gRNA control and kill switch strain. Cas9-expressing plasmids include a low (~5)-copy pSC101 origin of replication and gRNA-expressing plasmids include a medium (~20)-copy p15A origin of replication.**

| Strain name | Genetic parts | | | | | Knockouts | Figures |
|---|---|---|---|---|---|---|---|
| | gRNAs | cas9 | tetR | infA | tlpA | | |
| **No gRNA controls** | | | | | | | |
| Control | None | Plasmid | Unoptimized | None | None | None | Fig. 1c |
| Control | None | 4 Integrations | Unoptimized | None | None | None | Fig. 1h |
| Control | None | 4 Integrations | Unoptimized | Optimized | None | None | Fig. 3a |
| Control | None | 4 Integrations | Unoptimized | Optimized | None | ΔrecA ΔpolB ΔdinB ΔumuBC | Figs. 3a, 3i, 3j, 4b, 4e, 5b, 5e, 5f |
| **aTc-inducible kill switches** | | | | | | | |
| groL-1 | groL-1 | Plasmid | Unoptimized | None | None | None | Fig. 1c |
| groL-2 | groL-2 | Plasmid | Unoptimized | None | None | None | Fig. 1c |
| ile-1 | ile-1 | Plasmid | Unoptimized | None | None | None | Fig. 1c |
| ile-2 | ile-2 | Plasmid | Unoptimized | None | None | None | Fig. 1c |
| rrs-1 | rrs-1 | Plasmid | Unoptimized | None | None | None | Fig. 1c |
| rrs-2 | rrs-2 | Plasmid | Unoptimized | None | None | None | Fig. 1c |
| Int X1 | ile-2 | 1 Integration | Unoptimized | None | None | None | Fig. 1f |
| Int X2 | ile-2 | 2 Integrations | Unoptimized | None | None | None | Fig. 1f |
| Int X3 | ile-2 | 3 Integrations | Unoptimized | None | None | None | Fig. 1f |
| Int X4 | ile-2 | 4 Integrations | Unoptimized | None | None | None | Figs. 1f, 1h |
| Initial 2-gRNA | groL-2 and ile-2 | 4 Integrations | Unoptimized | None | None | None | Fig. 1h |
| Optimized 2-gRNA | Optimized groL-2 and ile-2 | 4 Integrations | Unoptimized | None | None | None | Figs. 1h, 2a |
| ABX-independent 2-gRNA | Optimized groL-2 and ile-2 | 4 Integrations | Unoptimized | Unoptimized | None | None | Fig. 2b |
| CRISPRks | Optimized groL-2, ile-2, and rrs-2 | 4 Integrations | Unoptimized | Optimized | None | None | Figs. 2c-f, 3a, 3e |
| CRISPRks Δrpdu | Optimized groL-2, ile-2, and rrs-2 | 4 Integrations | Unoptimized | Optimized | None | ΔrecA ΔpolB ΔdinB ΔumuBC | Figs. 3a, 3f, 3i, 3j |
| **2-input kill switches** | | | | | | | |
| Initial 2-input CRISPRks Δrpdu | Optimized groL-2, ile-2, and rrs-2 | 4 Integrations | Unoptimized | Optimized | Present | ΔrecA ΔpolB ΔdinB ΔumuBC | Fig. 4b |
| [Optimized] 2-input CRISPRks Δrpdu | Optimized groL-2, ile-2, and rrs-2 | 4 Integrations | Optimized | Optimized | Present | ΔrecA ΔpolB ΔdinB ΔumuBC | Figs. 4b, 4d, 4e, 5b, 5e, 5f |

units (CFUs) obtained in the non-permissive condition (+aTc) to CFUs obtained in the permissive condition (−aTc). Interestingly, the killing efficiencies for the multi-target gRNAs did not exceed the efficiencies of the single-target gRNAs. At least one gRNA for each target gene achieved a fraction viable of $10^{-4}$–$10^{-5}$. The kill switch rapidly triggered cell death, with maximum killing efficiencies detected after just 1.5 h of aTc induction (Supplementary Fig. 1a). However, efficiencies were significantly reduced by 2.5 h of induction, suggesting that a subpopulation of cells harboring inactive kill switches were able to repopulate.

**Improving kill switch stability through functional redundancy.** To determine the source of kill switch inactivation, we identified isolates exhibiting loss-of-function from the aTc-induction assay, and sequenced the cas9, gRNA, and tetR expression cassettes. While a small number of isolates harbored mutated gRNA ($10 ± 14\%$) and tetR ($3 ± 6\%$) cassettes, a large majority harbored mutated cas9 expression cassettes ($80 ± 19\%$), with mutations predominantly in the $P_{tet}$ promoters (Fig. 1d). To decrease the probability of Cas9 expression inactivation, we integrated functionally redundant $P_{tet}$-cas9 expression cassettes into four genomic neutral integration sites[11] (Fig. 1e). Each successive integration improved the efficiency of the ile-2 gRNA kill switch, with the four $P_{tet}$-cas9 integration strain (Int X4) achieving a fraction viable 10 times lower than the plasmid-based Cas9 expression strain (Fig. 1f). Chromosomal expression of four functionally redundant Cas9 cassettes did not alter the relative efficiency of the six gRNAs (Supplementary Fig. 1b).

Diminishing returns were observed by the fourth $P_{tet}$-cas9 integration, suggesting a shift in inactivation method. We sequenced non-functional Int X4 isolates and identified the promoter of the $P_{tet}$-gRNA expression cassette as the primary source of instability (Fig. 1g). Applying the same principle of functional redundancy, we combined the groL-2 and ile-2 gRNA expression cassettes onto one plasmid and achieved aTc-dependent cell death below the limit of detection after 1.5 h and a fraction viable below $10^{-6.6}$ (Fig. 1h). However, this initial 2-gRNA kill switch caused significant cell death even in the absence of aTc, limiting the therapeutic potential of the strain. To optimize gRNA expression and minimize leaky killing, we simultaneously mutagenized the -35 sites of the two gRNA $P_{tet}$ promoters and assayed colonies from the library (Supplementary Fig. 1c, d). We isolated one 2-gRNA kill switch that maintained full killing in the presence of aTc without leaky killing in the absence of aTc (Fig. 1h). The optimized 2-gRNA kill switch achieved a fraction viable of less than $10^{-8.6}$, surpassing the killing efficiency of $10^{-8}$ recommended by the NIH Office of Biotechnology Activities[38]. The optimized kill switch also displayed an improved aTc sensitivity relative to the single ile-2 gRNA kill switch (Supplementary Fig. 1e).

**Eliminating the reliance on antibiotics for efficient killing.** When the antibiotic (ABX) used to maintain the gRNA expression plasmid was removed from selection plates, we found the killing efficiencies of both 2-gRNA kill switches to be severely reduced due to loss of the gRNA plasmid during replication (Fig. 2a, Supplementary Fig. 2a). To enable stability of the kill

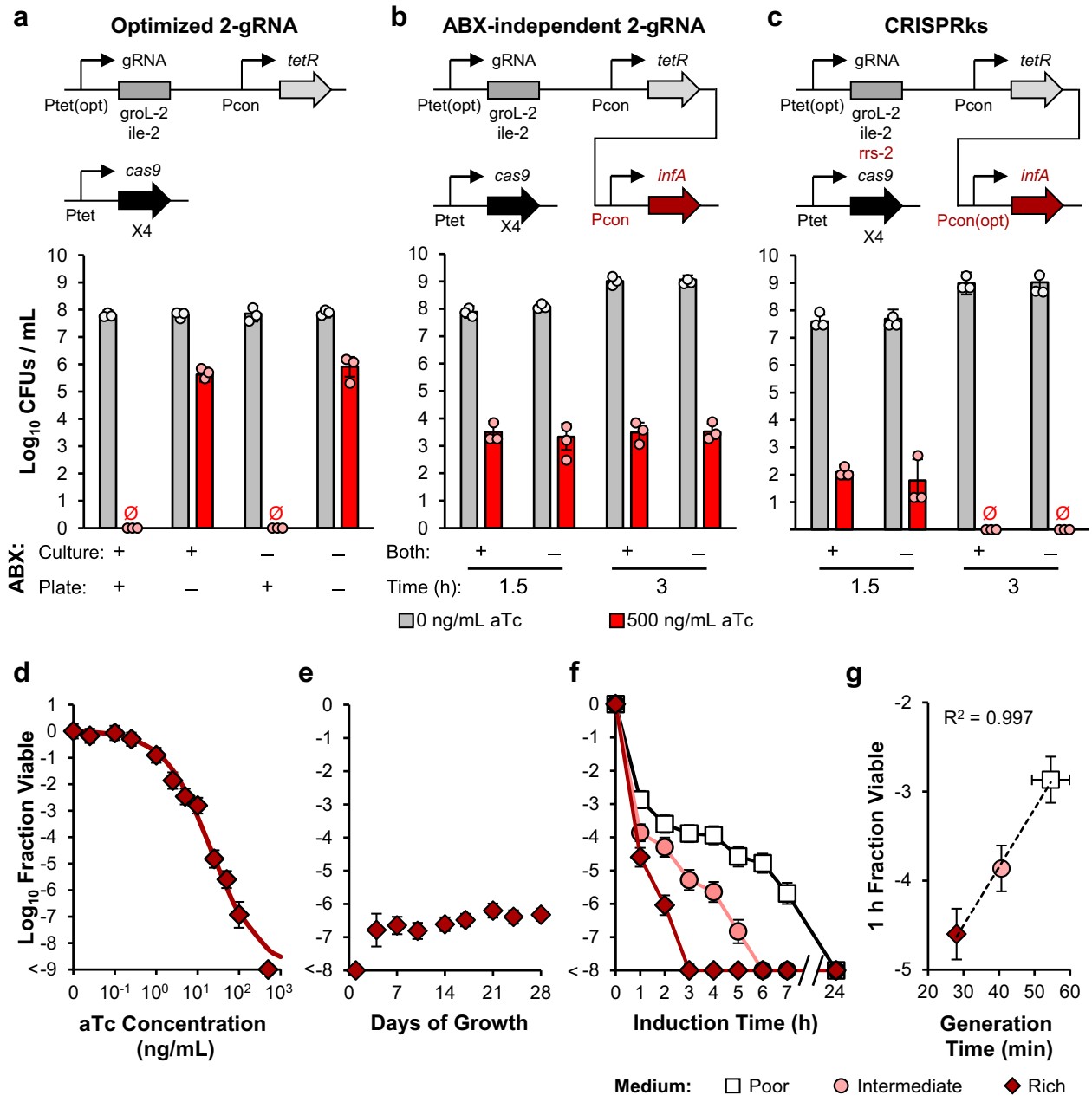

**Fig. 2 Eliminating the reliance on antibiotics for efficient killing. a–c** System schematics and log₁₀ CFU values for the following kill switch strains:
**a** optimized 2-gRNA circuit, which has four genome-integrated $P_{tet}$-*cas9* expression cassettes (X4) and optimized groL-2 and ile-2 $P_{tet}$-gRNA expression cassettes, **b** ABX-independent 2-gRNA circuit, which has four genome-integrated $P_{tet}$-*cas9* expression cassettes (X4), optimized groL-2 and ile-2 $P_{tet}$-gRNA expression cassettes, and an unoptimized constitutive *infA* expression cassette to complement a genomic *infA* knockout, and **c** CRISPRks, which has four genome-integrated $P_{tet}$-*cas9* expression cassettes (X4), optimized groL-2, ile-2, and rrs-2 $P_{tet}$-gRNA expression cassettes, and an optimized constitutive *infA* expression cassette to complement a genomic *infA* knockout. Exponential phase cells for each strain were induced with 0 and 500 ng/mL aTc for 1.5 h (**a**), and 1.5 or 3 h (**b** and **c**) in LB with and without spectinomycin. CFUs were determined by plating onto LB agar with and/or without spectinomycin. Differences between the circuit in a and the circuits in b and c are highlighted in the construct schematics in red. *gRNA*, *tetR*, and *infA* cassettes are located on the same plasmid (connected lines), while cas9 is located exclusively in the genome. 'Both' denotes whether antibiotics were present in both the liquid and solid phase media. **d** aTc-inducible killing transfer curve for the CRISPRks strain after 3 h of induction in LB without antibiotics. Points represent experimental data while the line represents the fitted curve. **e** Long-term stability assessment of the CRISPRks strain. Each day for 28 days, three replicates of the strain were diluted 250X into fresh LB without antibiotics and grown for 24 h. Every 3–4 days, exponential phase cells were induced with 0 and 500 ng/mL aTc for 3 h and plated on LB agar without antibiotics for CFU quantification. **f** Log₁₀ Fraction Viable of the CRISPRks strain in response to 500 ng/mL aTc in M9+0.4% glucose (poor), M9+0.4% glucose+0.2% casamino acids (intermediate), and LB (rich). **g** Correlation between generation time and fraction viable. Fraction viable values of <−8 or <−9 had no colonies obtained from cultures receiving aTc. Values and error bars are the average and standard deviation of biological triplicate, respectively. See also Supplementary Figs. 2, 3 and Supplementary Table 1. Source data are provided as a Source Data file.

switch plasmid constructs in the absence of antibiotics, we implemented a modified version of a previously described ABX-free plasmid maintenance method[39,40]. We expressed the essential gene *infA* (required for initiation of protein synthesis) using an intermediate strength constitutive promoter on the gRNA plasmid and knocked it out of the genome of the four $P_{tet}$-*cas9* integration strain. With this system, EcN needs to maintain the plasmid to provide InfA and survive. The antibiotic resistance gene was not removed from the plasmid to allow for CFU quantification in future experiments using mixed strains. Expressing InfA on the gRNA plasmid eliminated the dependence on ABX but lowered the efficiency of the kill switches (Fig. 2b, Supplementary Fig. 2b).

Switching a plasmid to InfA-based selection has been shown to affect plasmid copy number[39]. We therefore improved the efficiency of the ABX-independent 2-gRNA kill switches using two methods: increasing gRNA expression by adding an rrs-2 gRNA expression cassette library and tuning the strength of the constitutive *infA* cassette. We performed both optimization methods independently for both the ABX-independent initial 2-gRNA and optimized 2-gRNA kill switches (Supplementary Fig. 2c–f). Both methods improved kill switch efficiency but failed to restore complete killing. However, when we incorporated a third gRNA expression cassette into the InfA-optimized, optimized 2-gRNA kill switch, we obtained two variants that achieved complete killing in the presence of aTc with uninhibited growth in the absence of aTc (Supplementary Fig. 3a, b). To identify the superior CRISPRks variant, we assayed each for their aTc-response time, aTc-sensitivity, and long-term stability (Fig. 2c, Supplementary Fig. 3c–f). Both variants were inactivated at similarly low rates, with a small population of non-functional cells appearing after 14 days (112 generations) of growth in vitro. However, the Mut14 variant displayed a significantly better response time, aTc sensitivity, and long-term killing efficiency. The Mut14 CRISPRks achieved complete killing by 3 h of aTc induction, responded to aTc with a half-maximal concentration of 21 ng/mL, and displayed highly stable killing over 28 days of growth (Fig. 2c–e). In addition, the kill switch reliably achieved complete killing of EcN in diverse nutrient conditions, with a linear correlation between generation time and log fraction viable at 1 h (Fig. 2f, g).

**Complete killing is achieved by the CRISPRks in vivo after knocking out key components of the SOS response and providing intra-niche competition.** Having demonstrated the dependence of the kill switch on growth conditions, we next sought to predict kill switch efficacy in vivo. We developed an in vitro assay that more closely mirrors the gut, a complex environment with lower levels of oxygen, nutrients, and mixing than standard in vitro conditions. When induced with aTc in minimal media, micro-aerobically, and without shaking, the CRISPRks (circuit schematic in Fig. 2c) achieved a fraction viable of $10^{-3.3}$ after 24 h of induction (Fig. 3a). After induction, we detected a large fraction of surviving colonies with kill switch inactivation (Fig. 3b). By 48 h of induction, the CRISPRks monoculture had rebounded to pre-induction levels (Fig. 3a). We hypothesized that supplementing the assay with microbes that fill the same niche as the CRISPRks strain could prevent the rapid regrowth of escape mutants and maximize killing efficiency[31]. We introduced competition into the assay by incubating the spectinomycin-resistant CRISPRks strain at a 1:1 ratio with a chloramphenicol-resistant control strain. In this mixture, the CRISPRks strain achieved an improved fraction viable of $10^{-4.2}$ after 24 h of aTc induction (Fig. 3a, Supplementary Fig. 4a). Although a population of the 1:1 consortium cells with non-functional kill switches was detected among the colonies that

survived aTc induction, it was significantly smaller than the population detected from the CRISPRks monocultures (Fig. 3b). Importantly, the fraction viable remained stable over the course of the experiment, suggesting that intra-niche competition is important to prevent the repopulation of escape mutants.

CRISPRks escape mutants were observed in the assay emulating in vivo conditions but not when tested under more optimal conditions (Fig. 2f), suggesting that the mutations accumulated de novo during the assay rather than existed in the inoculum population. Single DNA DSBs, as induced by CRISPR-Cas9, have been shown to strongly induce the SOS response in *E. coli*[41,42]. This response increases the mutation rate of the cell through the expression of recombinase genes, including *recA*, and diverse error-prone DNA polymerase genes, including *polB*, *dinB*, and *umuDC*. Thus, we hypothesized that a nutrient- and growth-limited environment, which have been shown to reduce per-cell-protein production[43], reduces expression of the kill switch, impairing the DNA-cleavage rate and allowing the survival of daughter cells with an induced SOS response and elevated kill switch inactivation rate. To reduce SOS-response-mediated DNA mutagenesis, we knocked out *recA*, *polB*, *dinB*, and *umuDC* (Δrpdu) from the CRISPRks strain. The CRISPRks Δrpdu kill switch maintained complete killing in optimal growth conditions (Supplementary Fig. 4b). In an assay emulating in vivo conditions, the CRISPRks Δrpdu strain achieved a similar fraction viable as the CRISPRks strain in the absence of competition (Fig. 3a). However, CRISPRks Δrpdu had a significantly smaller population of non-functional survivors after 24 h of induction, indicating a reduced rate of inactivation, but with a similar mutation profile (Fig. 3b, c). Most mutations for both strains were in the promoters of the $P_{tet}$-*cas9* and $P_{tet}$-gRNA expression cassettes. No mutations were identified in the *infA* cassette or the gRNA target sites. Strikingly, in the presence of competition from the control strain, CRISPRks Δrpdu cells were eliminated from the culture by 72 h of aTc induction (Fig. 3a, Supplementary Fig. 4a). Using the same assay, we found that the CRISPRks Δrpdu strain was able to be eliminated from the culture at control:kill switch ratios as low as 1:1000 (Supplementary Fig. 4c). This is important because it suggests the mitigation of kill switch escape afforded by intra-niche competition is robust against stochastic variation in sample preparation and colonization efficiencies of the two strains; further, a lower proportion of control relative to engineered probiotic would help maximize therapeutic potential. Both the CRISPRks and CRISPRks Δrpdu strains showed similar long-term killing efficiencies, kill switch inactivation rates, and mutation patterns in optimal growth conditions (Fig. 4d–f). As such, while the Δrpdu knockouts significantly reduce induction-dependent kill switch inactivation, they have no impact on the mutation rate during normal DNA replication.

We then tested the efficacy of the CRISPRks and CRISPRks Δrpdu strains in vivo. C57BL/6 mice were treated with streptomycin to enable EcN colonization[44] (Fig. 3d). 24 h later, $10^8$ CFU of kill switch or control EcN, with or without the Δrpdu knockouts, were delivered to mice by oral gavage. 24 hours after EcN gavage, mice were switched to aTc treatment water ($10^5$ ng/mL aTc + 5% sucrose) or control water (5% sucrose) *ad libitum*, and fecal samples were collected longitudinally (Fig. 3d). While both kill switch strains demonstrated significant killing activity in vivo, the CRISPRks Δrpdu strain exhibited improved killing efficacy with a 4-log reduction in fecal titers after 24 hours of aTc treatment, compared to a 1-log reduction for the CRISPRks strain at the same timepoint (Fig. 3e, f). In addition, as observed in vitro (Fig. 3b), the Δrpdu knockouts mitigated the incidence of escape mutants; fewer CRISPRks Δrpdu isolates (57 ± 37% versus 100 ± 0%) recovered from stool at the 24-hour timepoint

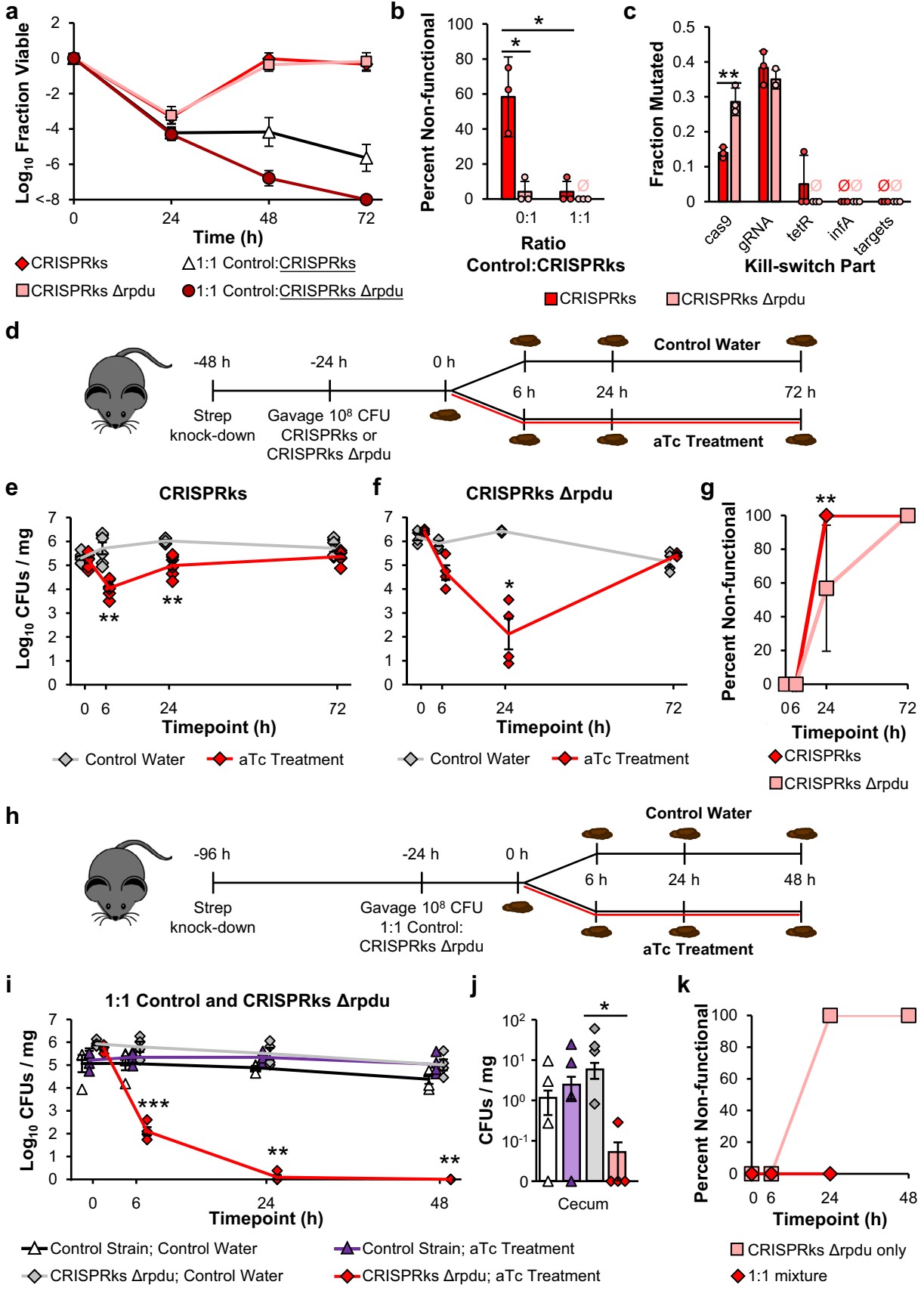

**Fig. 3 Complete killing is achieved by the CRISPRks in vivo after knocking out key components of the SOS response and providing intra-niche competition. a** $Log_{10}$ Fraction Viable of the CRISPRks and CRISPRks $\Delta$rpdu kill switch strains when incubated alone or at a 1:1 ratio with the no gRNA control. Cells were cultured anaerobically at 37 °C without shaking in M9+0.4% glucose and induced with 0 or 500 ng/mL aTc. Fraction viable values of <-8 had no colonies obtained from cultures receiving aTc. Values and error bars are the average and standard deviation of biological triplicate, respectively. **b** Percent of cells that survived 24 h of aTc induction with non-functional kill switches. 24 colonies (8 from each replicate) of each condition were tested for functionality. Values and error bars are the average and standard deviation, respectively. **c** Sequencing results from 24 CRISPRks and 24 CRISPRks $\Delta$rpdu survivors (8 from each replicate) from **a** with non-functional ile-2 kill switches. The fraction mutated is the fraction of total sequenced cassettes that contained a mutation. Values and error bars are the average and standard deviation, respectively. **d** Schematic for testing the kill switches in vivo. 24 h after a streptomycin treatment, $10^8$ CFUs of CRISPRks or CRISPRks $\Delta$rpdu were gavaged into C57BL/6 mice. 24 h after gavage, mice receiving each kill switch strain were split into two groups. One group received control water (5% sucrose) and the other group received aTc treatment water ($10^5$ ng/mL aTc + 5% sucrose). Fecal samples were collected 0, 6, 24, and 72 h after gavage for CFU quantification. **e, f** $Log_{10}$ CFUs/mg feces of **e** CRISPRks or **f** CRISPRks $\Delta$rpdu strains from mice receiving control water or aTc treatment water. Points are the average of two technical replicates. Lines and error bars are the average and standard error from 6 (**e**) or 4 (**f**) mice across two cages, respectively. **g** Percent of CRISPRks and CRISPRks $\Delta$rpdu cells that survived aTc treatment with non-functional kill switches. To assay for kill switch functionality, exponential phase cells were induced with 0 and 500 ng/mL aTc for 3 h and the absorbance at 600 nm quantified. Induced to uninduced absorbance ratios within three standard deviations of the no gRNA control strain were deemed non-functional. 24 colonies (12 from each cage) at each timepoint were tested for functionality. Values and error bars are the average and standard deviation, respectively. **h** Schematic for testing control and kill switch co-gavage in vivo. 72 h after a streptomycin treatment, $10^8$ CFUs of a 1:1 ratio mixture of the no gRNA control strain and CRISPRks $\Delta$rpdu were gavaged into C57BL/6 mice. 24 h after gavage, mice receiving each kill switch strain were split into two groups. One group received control water (5% sucrose) and the other group received aTc treatment water ($10^5$ ng/mL aTc + 5% sucrose). Fecal samples were collected 0, 6, 24, and 48 h after gavage for CFU quantification. At the conclusion of the experiment (192 h), mice were sacrificed and cecal contents were collected for CFU quantification. **i, j** $Log_{10}$ CFUs/mg feces (**i**) or cecal contents (**j**) of CRISPRks $\Delta$rpdu from mice receiving a 1:1 ratio mixture of CRISPRks $\Delta$rpdu and the no gRNA control strain. Points are the average of two technical replicates. Cecal contents were sampled 8 days after gavage. Lines and error bars are the average and standard error from 4 mice across two cages, respectively. **k** Percent of CRISPRks $\Delta$rpdu cells that survived aTc treatment with non-functional kill switches when gavaged alone or in a 1:1 ratio mixture with the no gRNA control strain. 24 colonies (12 from each cage) at each timepoint were tested for functionality. Statistical comparisons were performed using two-tailed unpaired $t$-tests (**b** and **c**) or two-tailed mixed model ANOVA with Sidak's multiple comparisons (**e**–**g**, **i**, and **j**), (*$P < 0.05$; **$P < 0.01$; ***$P < 0.001$). See also Supplementary Figs. 4 and 5. Source data with p-values are provided as a Source Data file.

exhibited loss of kill switch activity in follow-up in vitro assays (Fig. 3g).

CRISPRks-mediated reduction of EcN titers in vivo was transient even with the $\Delta$rpdu knockouts. Fecal kill switch titers in aTc-treated mice approached that of mice given control water by 72 hours of treatment (Fig. 3e, f). At this timepoint, 100% of recovered kill switch isolates from aTc-treated mice demonstrated loss-of-function in follow-up in vitro assays (Fig. 3g), indicating a bloom of an escape mutant population in vivo. We additionally observed that fecal titers of the control strains were low on average and highly variable at treatment baseline (Supplementary Fig. 4g, h). In contrast, fecal titers of CRISPRks strains consistently reached 5-6 logs at treatment baseline (Fig. 3e, f). We hypothesized that the spectinomycin-resistant kill switch strains were cross-resistant to the streptomycin used for knock-down of the native microbiota, a phenotype not afforded to the chloramphenicol-resistant control strains. Indeed, we observed that baseline fecal titers of chloramphenicol-resistant EcN were dependent on the length of the interval between streptomycin and EcN gavage, while those of an isogenic spectinomycin-resistant strain of EcN were not (Supplementary Fig. 5a, b). We sought to optimize the microbiota knockdown protocol to consistently achieve equal baseline titers of the control and CRISPRks strains, either by extending the interval between streptomycin treatment and EcN gavage or through use of an alternative antibiotic, and determined that waiting 72 h after streptomycin treatment enables equivalent baseline titers of chloramphenicol-resistant and spectinomycin-resistant EcN (Supplementary Fig. 5b). In contrast, omission of any antibiotic treatment resulted in undetectable levels of either EcN strain (Supplementary Fig. 5c), indicating a failure to colonize, while treatment with carbenicillin[45] 48 hours before EcN gavage resulted in low and variable titers of both EcN strains (Supplementary Fig. 5d).

We therefore carried out another experiment in mice with the CRISPRks $\Delta$rpdu and control $\Delta$rpdu strains with a 72 h interval between streptomycin and EcN gavage. Motivated by the hypothesis that intra-niche competition by an isogenic non-kill switch EcN strain can support eradication of the kill switch EcN population, we included an additional arm in which we co-gavaged the kill switch and control strains at a 1:1 ratio (Fig. 3h). This is an attractive approach when the goal is eradication of a specific population of engineered microbes, rather than a probiotic species itself, from the gut. Using the 72 h interval, we were able to achieve equivalent baseline titers of spectinomycin-resistant CRISPRks $\Delta$rpdu and chloramphenicol-resistant control $\Delta$rpdu EcN (Fig. 3i, Supplementary Fig. 5e, f). Remarkably, when CRISPRks $\Delta$rpdu EcN was co-gavaged with control $\Delta$rpdu EcN, the CRISPRks $\Delta$rpdu strain was no longer detectable in stool by 48 hours of aTc treatment (Fig. 3i). Except for a single colony observed after plating undiluted fecal homogenate, the CRISPRks $\Delta$rpdu strain remained undetectable in cecal contents after 8 days of aTc treatment, demonstrating almost complete eradication of the engineered microbe from the mice (Fig. 3j). In contrast, titers of control $\Delta$rpdu EcN in the same mice remained stable over the course of aTc treatment, as did titers of both strains (similarly co-gavaged) in mice treated with control water (Fig. 3i). Correspondingly, we were still able to detect EcN in the ceca of these mice at sacrifice, with cecal titers not significantly different between the control strain in either treatment arm and the CRISPRks $\Delta$rpdu strain in the control water arm (Fig. 3j). After 24 hours of aTc treatment in vivo, 100% of CRISPRks $\Delta$rpdu isolates recovered from mice colonized with only this strain exhibited loss of kill switch function in vitro, while no loss of function was observed in isolates of the same strain from co-gavaged mice (Fig. 3k). These data indicate that a combined approach of kill switch induction and intra-niche competition can mitigate the emergence of escape mutant populations in vivo.

**Development of a 2-input CRISPRks that responds to both aTc and reduced temperatures.** Several temperature sensors have been characterized in other strains of *E. coli*[30,46,47]. Of these

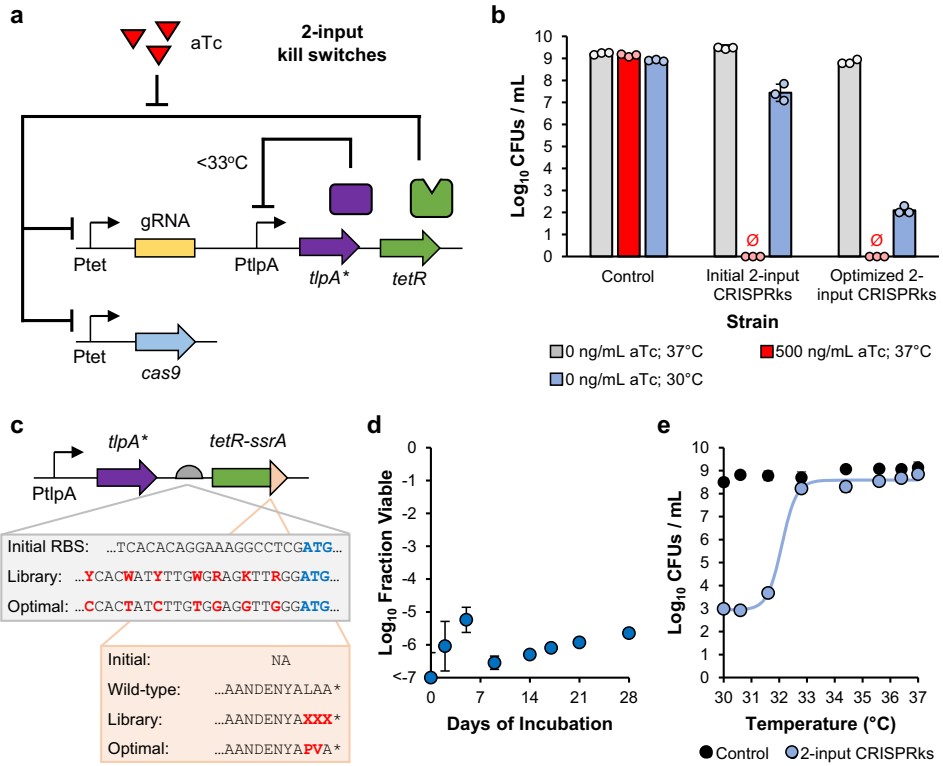

**Fig. 4 Development of a 2-input CRISPRks that responds to both aTc and reduced temperatures. a** Schematic of the joint aTc- and temperature-inducible CRISPRks system. Cas9 is expressed from four genome-integrated aTc-inducible $P_{tet}$-$cas9$ cassettes, and gRNAs are expressed from three aTc-inducible $P_{tet}$-gRNA expression cassettes on a plasmid. TetR, which regulates the expression of the $P_{tet}$ promoters in an aTc-dependent manner, is expressed by the $P_{tlpA}$ promoter on the gRNA plasmid. Activity of the $P_{tlpA}$ promoter is regulated by the TlpA* transcription factor. TlpA* is expressed from the same $P_{tlpA}$ promoter and regulates its own expression in a negative feedback loop. In the presence of aTc or at temperatures less than 33 °C, the Cas9-gRNA complex is produced, leading to cell death. **b** $Log_{10}$ CFUs for the no gRNA control, initial 2-input CRISPRks, and optimized 2-input CRISPRks. Exponential phase cells for each strain were induced with 0 and 500 ng/mL aTc at 37 °C for 3 h. Cultures were then plated on LB agar without antibiotics and incubated overnight at 37 °C (both 0 and 500 n/mL aTc cultures) or for seven days at 30 °C (0 ng/mL cultures) for CFU quantification. **c** The expression and stability of TetR was optimized for the 2-input CRISPRks by simultaneously tuning the strength of the RBS and inserting a C-terminus SsrA degradation tag library onto TetR. **d** Long-term stability of the optimized 2-input kill switch. Each day for 28 days, three replicates of the strain were diluted 250X into fresh LB without antibiotics and grown for 24 h. Every 3-4 days, exponential phase cells were plated on LB agar without antibiotics and incubated overnight at 37 °C and for 7 days at 30 °C for CFU quantification. Fraction viable values of <-7 had no colonies obtained from cultures receiving aTc. **e** Cell death transfer curve with respect to temperature for the no gRNA control and the optimized 2-input kill switch. Exponential phase cells were incubated in a thermocycler for 5 h at a range of temperatures: 30, 30.6, 31.6, 32.8, 34.4, 35.6, 36.4, and 37 °C. Points represent experimental data while the line represents the fitted curve. Values and error bars are the average and standard deviation of biological triplicate, respectively. See also Supplementary Fig. 6 and Supplementary Table 1. Source data are provided as a Source Data file.

sensors, the $P_{tlpA}$ promoter with a modified TlpA regulator protein from *Salmonella* demonstrated the highest fold-change in expression in response to a temperature reduction. TlpA is a transcriptional regulator that assembles into homodimers and represses transcription from the $P_{tlpA}$ promoter at low temperatures[48]. At high temperatures, the dimers are unable to form, allowing transcription. To characterize the sensor in EcN, we obtained the $P_{tlpA}$-$tlpA$ temperature sensing system from *Salmonella typhimurium* SL1344 genomic DNA and used it to drive expression of a green fluorescent protein (GFP) reporter. The wild-type TlpA sensor induced GFP expression with a half maximal fluorescence at 43.4 °C, significantly above our desired range (Supplementary Fig. 6a). We next inserted five amino acid substitutions identified by Pirner et al. to generate the modified TlpA sensor, TlpA*[30]. As previously demonstrated, the mutations shifted the half maximal fluorescence to a temperature of 35.6 °C (Supplementary Fig. 6b). Expression of GFP was 97% repressed at a temperature of 34 °C compared to 37 °C.

To make a 2-input CRISPRks that induces cell death in response to both aTc and a temperature downshift (i.e., an OR gate), we replaced the constitutive promoter driving TetR expression with the $P_{tlpA}$-$tlpA$* temperature sensor (Fig. 4a). With this design, TetR expression is inhibited by TlpA* at low temperatures (<33 °C), de-repressing expression of the kill switch. At 37 °C, TetR is expressed, and the kill switch remains sensitive to aTc. This initial 2-input CRISPRks achieved complete cell death in response to aTc, but only a modest reduction in fraction viable ($10^{-1.7}$) at 30 °C (Fig. 4b). We hypothesized that the temperature-dependent response was poor because the circuit inhibits new TetR production but not the activity of TetR already in the cell. Numerous cell divisions may be required to dilute cytosolic TetR concentrations below the threshold for kill switch induction. To test this hypothesis, we performed a liquid-phase, temperature-response kill switch assay using a range of initial dilution factors, where larger starting dilutions would allow more growth and smaller per-cell TetR concentrations. The killing

efficiency was directly correlated to the dilution factor, confirming the dependence on cell divisions for kill switch activation (Supplementary Fig. 6c). To improve the response to temperature, we sought to uncouple TetR removal from cell growth. We simultaneously optimized the expression level and stability of TetR by inserting a ribosome binding site (RBS) library and a C-terminal SsrA degradation tag library, respectively (Fig. 4c). We tested over 500 2-input CRISPRks Δrpdu variants from the combined RBS and degradation tag library and identified eight that achieved efficient cell death at 30 °C with minimal growth inhibition at 37 °C (Supplementary Fig. 6d). To select the best kill switch, we assayed select variants for long-term stability and temperature sensitivity (Supplementary Fig. 6e, f). The final selected variant was highly stable over 28 days of growth (Fig. 4d) and demonstrated an ultrasensitive response to temperature, with no killing at 33-37 °C and strong killing at temperatures less than 32 °C (Fig. 4e).

**The 2-input CRISPRks efficiently kills EcN in response to both aTc treatment and excretion from mice.** We next tested the in vivo efficiency of the 2-input CRISPRks Δrpdu strain when gavaged into streptomycin-treated mice alone, or in co-gavage with the cognate control strain at a 1:1 ratio (Fig. 5a). In a control arm, mice were instead singly-gavaged with the control strain as before. To assay chemical induction of the 2-input kill switch, 24 hours after EcN gavage, mice were switched to aTc treatment water or control water *ad libitum*, and fecal samples were collected longitudinally over a week (Fig. 5a). To assay temperature induction of the 2-input kill switch, collected stool was processed at both 37 °C and room temperature (22 °C); results from samples processed at 37 °C reflect only within-gut aTc induction of kill switch activity, while those processed at 22 °C additionally reflect temperature induction of the kill switch after excretion.

When singly-gavaged, aTc induction of the 2-input CRISPRks Δrpdu strain within mice resulted in a significant (3-log) reduction in fecal titers. However, this response was again transient over the first 48 h of treatment with the kill switch population rebounding to baseline levels by 72 h and remaining high in stool on day 7 and in the ceca on day 8 (Supplementary Fig. 7b, c, top). Temperature induction of the 2-input CRISPRks Δrpdu strain in stool collected from these mice greatly improved killing efficacy; we consistently observed 1- to 2-log CFU/mg EcN in stool from mice that were not induced with aTc, 5-log lower than control strain titers, indicating that this reduction was driven by temperature induction alone (Supplementary Fig. 7b, bottom). When combined with aTc induction in mice, temperature induction (i.e., post-excretion) further reduced fecal titers such that we were unable to detect EcN in stool from these mice at the 24- and 48-h timepoints (Supplementary Fig. 7b, bottom). However, the kill switch population was not completely eradicated as we were able to detect it in stool on day 7 and in the ceca on day 8 (Supplementary Fig. 7b, c, bottom). Interestingly, we also observed greater variability in stool titers of the control strain in mice treated with aTc compared to control water, including one mouse in which we were unable to detect control strain at multiple timepoints (Supplementary Fig. 7a). This high variability may be due to aTc induction of the Cas9 complex (which the control strain carries despite lacking gRNAs), leading to decreased in vivo fitness, potentially coupled with a rise of control strain subpopulations with inactivating mutations (e.g., mutations in the $P_{tet}$ promoter driving Cas9 expression).

Co-gavage of the 2-input CRIPRSks Δrpdu strain with the control strain, providing intra-niche competition, mitigated the bloom of an escape mutant population during aTc treatment (Fig. 5b, top). This was evidenced by significantly lower ($p = 0.0006$; mixed model ANOVA) stool and cecal titers of the kill switch strain in aTc-treated mice compared to control strain titers in the same mice, or to either strain in mice treated with control water. Correspondingly, fewer kill switch isolates recovered from co-gavaged mice exhibited loss of function in follow-up in vitro kill switch assays compared to isolates from singly-gavaged mice ($29 \pm 6\%$ compared to $92 \pm 12\%$ on day 3, and $67 \pm 24\%$ compared to $100\%$ by day 7) (Fig. 5c). As in the single-gavage arm (Supplementary Fig. 7), in samples from co-gavaged mice, temperature induction alone of the 2-input kill switch resulted in high killing efficacy (1-log CFU/mg feces [gray circles] compared to 4-log CFU/mg for the control strain in the same mice [white triangles]), but the kill switch population increased in titer by day 7 (Fig. 5b, bottom). When we sequenced non-functional isolates, we identified mutations throughout the *cas9*, gRNA, *tetR*, and *tlpA* expression cassettes, suggesting the presence of diverse inactivation mechanisms and that the stability of the kill switch cannot be easily improved through additional functional redundancies (Fig. 5d). Again, no mutations were identified in the *infA* cassette or the gRNA target sites.

Strikingly, a combination approach, including provision of intra-niche competition by a competing strain, within-gut chemical kill switch induction, and temperature kill switch induction outside the gut, was effective for eradicating the kill switch population; with these conditions, we were unable to detect the 2-input CRISPRks strain in stool beginning at the 24-hour timepoint and through the end of the experiment, nor in the ceca on day 8 (Fig. 5b, bottom [red circles]). In addition to plating directly from stool and ceca to quantify EcN, we used fecal and cecal samples to inoculate room temperature kinetic growth assays in rich media as a more sensitive measure of temperature-induced biocontainment, where any EcN population that may have been below the limit of detection of the direct plating assays has the opportunity to amplify (Fig. 5e, f, Supplementary Fig. 7d, e). In line with the direct plating results, while on days 7 and 8 (cecal timepoint), we observed some growth of the 2-input CRISPRks Δrpdu strain in cultures inoculated from mice not treated with aTc (gray lines), we observed no growth over 24 hours of the kill switch strain in cultures inoculated from mice that had been treated with aTc (red lines, Fig. 5e); with the exception of an average of $10^{1.4}$ kill switch colonies observed at the 72-hour timepoint, plating from the kinetic growth assays at their termination to quantify CFUs confirmed this result (Fig. 5f). In contrast, in growth assays inoculated from mice treated with aTc in the single-gavage arms, we observed no significant difference in titers of 2-input kill switch strains and control strains (Supplementary Fig. 7d, e). These data indicate that intra-niche strain competition is required to mitigate aTc-mediated selection for kill switch mutants that escape both chemical and temperature induction modalities.

It is important that biocontainment systems minimally impact the protein production capability and therapeutic potential of the engineered microbe. To determine how the kill switches designed here impact protein expression in EcN, we quantified the expression of a constitutively expressed GFP reporter using both a genome-integrated and plasmid-based cassette. Both the CRISPRks Δrpdu strain and the 2-input CRISPRks Δrpdu strain achieved GFP expression levels equivalent to wild-type EcN and the no gRNA control strain (Supplementary Fig. 8). This trend remained true for both genome-integrated and plasmid-based expression methods. As such, the aTc-responsive CRISPRks Δrpdu kill switch and the aTc- and temperature-responsive 2-input CRISPRks Δrpdu kill switch can be effectively applied towards the biocontainment of engineered therapeutic and diagnostic microbes.

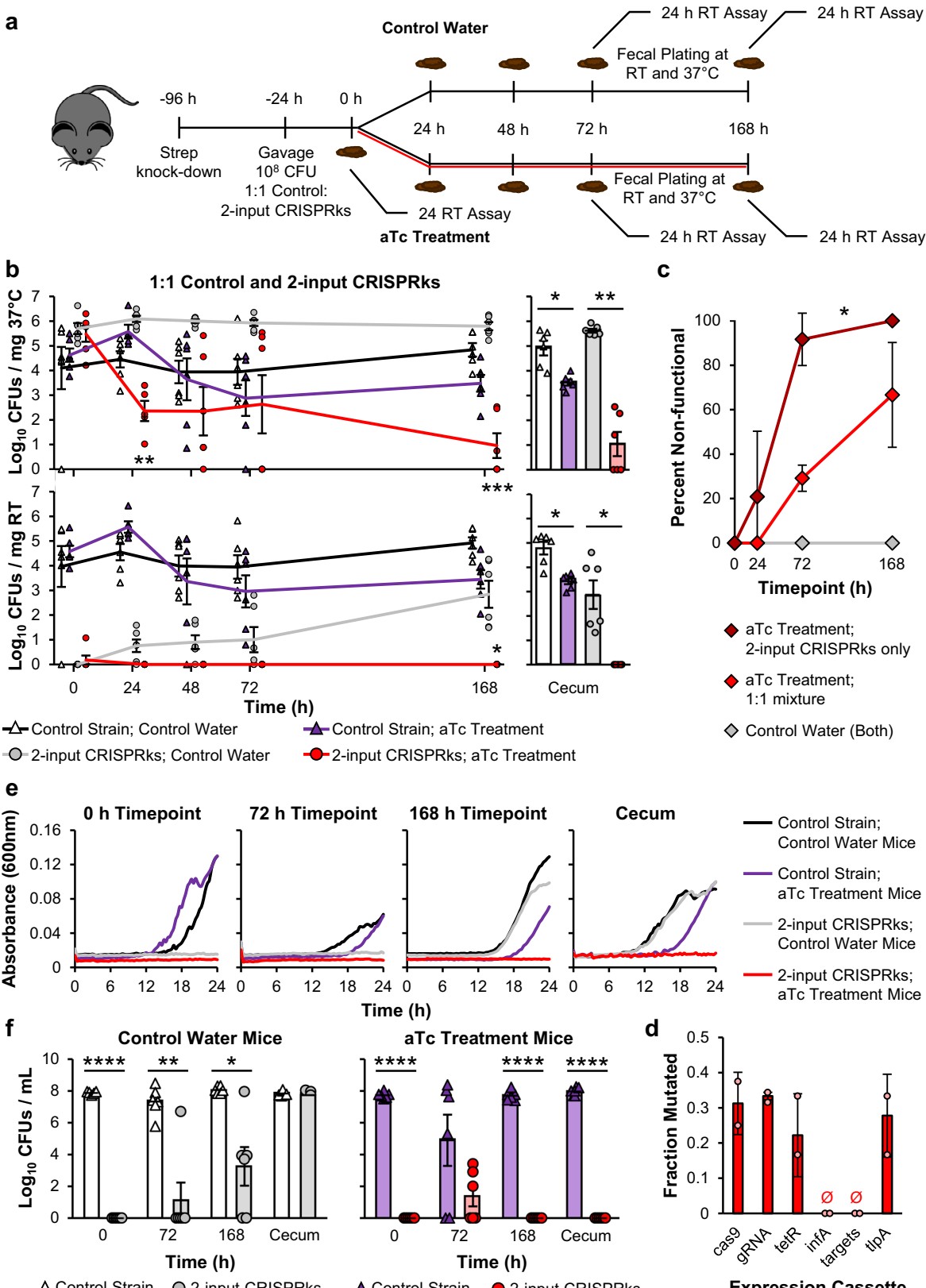

## Discussion

Robust control of the viability of engineered probiotics is essential for the host safety and the environmental protection. The aTc-only and 2-input CRISPRks strains developed here allow the growth of EcN to be tightly controlled during in vivo applications.

To develop the two CRISPRks strains, we explored a variety of methods for optimizing genetic stability, including incorporating multiple functionally redundant Cas9 and gRNA expression cassettes (Figs. 1 and 2), eliminating reliance on antibiotics for kill switch maintenance (Fig. 2), knocking out key SOS response

**Fig. 5 The 2-input CRISPRks efficiently kills EcN in response to both aTc treatment and excretion from mice. a** Schematic for testing co-gavage of the control strain and 2-input CRISPRks in vivo. 72 h after a streptomycin treatment, $10^8$ CFUs of a 1:1 ratio mixture of the no gRNA control strain and 2-input CRISPRks were gavaged into C57BL/6 mice. 24 h after gavage, mice were split into two groups. One group received control water (5% sucrose) and the other group received aTc treatment water ($10^5$ ng/mL aTc + 5% sucrose). Fecal samples were collected 0, 24, 48, 72, and 168 h after gavage for CFU quantification. At the conclusion of the experiment (192 h), mice were sacrificed and cecal contents were collected for CFU quantification. Fecal and cecal contents were plated at 37 °C overnight or room temperature (RT) for 48 h. Samples were also incubated in LB broth at RT for 24 h and plated at RT for 48 h for CFU quantification. **b** $Log_{10}$ CFUs/mg feces (left) or cecal contents (right) of 2-input CRISPRks cells from mice receiving a 1:1 ratio mixture of 2-input CRISPRks and the no gRNA control strain. CFUs were quantified at 37 °C (top) or RT (bottom). Points are the average of two technical replicates. Lines and error bars are the average and standard error from 6 mice across two cages, respectively. Statistical comparisons were made between control water and aTc treatment (top) or control strain and 2-input CRISPRks (bottom). **c** Percent of 2-input CRISPRks cells that survived aTc treatment or control water with non-functional kill switches when gavaged alone (from Supplementary Fig. 7b, top) or in a 1:1 ratio mixture of 2-input CRISPRks and the no gRNA control strain (from Fig. 5b, top). To assay for kill switch functionality, exponential phase cells were induced with 0 and 500 ng/mL aTc for 3 h and the absorbance at 600 nm quantified. Induced to uninduced absorbance ratios within three standard deviations of the no gRNA control strain were deemed non-functional. 24 colonies (12 from each cage) at each timepoint were tested for functionality through an in vitro aTc-response assay. **d** Sequencing results from 24 2-input CRISPRks survivors from Fig. 5b, top with non-functional kill switches. The fraction mutated is the fraction of total sequenced cassettes that contained a mutation. Mutations in the $P_{tlpA}$ promoter are counted as mutations in both the *tetR* and *tlpA* expression cassettes. **e** RT growth assays for control and 2-input CRISPRks cells obtained from fecal or cecal samples at different timepoints in Fig. 5b. **f** CFUs of control and 2-input CRISPRks cells from Fig. 5e following 24 h of RT growth. Points are the average of two technical replicates. Values and error bars are the average and standard error from 6 mice across two cages, respectively. Statistical comparisons were performed using two-tailed mixed model ANOVA (**c**) or two-tailed mixed model ANOVA with Sidak's multiple comparisons (**b** and **f**) (*$P < 0.05$; **$P < 0.01$; ***$P < 0.001$, ****$P < 0.0001$). See also Supplementary Fig. 7. Source data with p-values are provided as a Source Data file.

genes involved in DNA mutagenesis (Fig. 3), providing intra-niche competition (Figs. 3 and 5), and combining two layered methods of viability control (Figs. 4 and 5).

In this work, we addressed two routes of DNA mutagenesis that contributed to kill switch instability (Fig. 6a). First, we mitigated stochastic inactivation of kill switch constructs resulting from the natural and slow accumulation of errors that occurs during DNA replication[49,50]. Introduction of functionally redundant Cas9 and gRNA expression cassettes reduced the kill switch inactivation rate due to these processes, enabling kill switch stability for at least 28 days of growth. However, we continued to detect inactivated CRISPRks variants after about 14 days of growth (Supplementary Fig. 4d–f). The rate of evolution can potentially be further reduced by optimizing the metabolic burden required to maintain the circuit[51], improving the accuracy of DNA replication[52], and knocking out transposable elements[53,54]. The second route of kill switch inactivation involved SOS-mediated DNA mutagenesis in response to the DSBs caused by Cas9 (Fig. 6a). Escape mutants of our CRISPRks strain only arose via this process when tested in vivo or with in vivo-like conditions where resources and growth are limited[55]. In these conditions, limited induction of the kill switch may lead to incomplete cleavage of the multi-copy repli-cating chromosome. Low levels of DSBs per cell would allow daughter cells to survive with intact copies of the chromosome, but with an activated SOS response and an elevated rate of DNA mutagenesis. Expression of a genome-integrated GFP reporter is significantly weaker in minimal medium compared to rich medium, suggesting lower Cas9 expression levels in nutrient-poor conditions (Supplementary Fig. 8). In addition, even single double strand breaks have been shown to strongly induce the SOS response[41]. We successfully reduced SOS-mediated inactivation through the Δrpdu knockouts, allowing for complete killing of the CRISPRks strain in growth-limited conditions. Knocking out alternative recombinases with known low levels of activity may further improve the stability of the 2-input CRISPRks in vivo[56]. In addition, our current gRNAs target the chromosome near the origin of replication where growth-dependent copy numbers would be highest (Fig. 1b). Utilizing gRNAs that instead target the region where growth-dependent chromosomal copy numbers are lower may decrease the number of daughter cells that escape kill switch induction with intact genomes and reduce the potential for SOS-mediated kill switch inactivation.

Mutations in the $P_{tet}$-*cas9* and $P_{tet}$-gRNA cassettes each con-tributed to ~45% of the total observed mutations, highlighting the importance of functionally redundant expression cassettes (Fig. 6b). Importantly, the $P_{tet}$ promoters within these cassettes were the primary source of instability. The $P_{tet}$ promoter used in these kill switches contains two identical 19 bp *tet* operator sites. Over 50% of the total observed mutations were 25 bp deletions including one of these operators and most of the -35 site (Fig. 6c). These deletions can occur through RecA-dependent homologous recombination, or RecA-independent rearrangement of tandem repeat sequences by replication slippage, sister-chromosome exchange-associated slippage, and single-strand annealing[50]. Replacing the $P_{tet}$ promoters with engineered variants that have only one *tet* operator site or operator sites with lower sequence identity, or using alternative chemical-inducible promoters lack-ing internal homology, may further improve the stability of the kill switches.

Biocontainment of genetically engineered organisms must not neglect biocontainment of the corresponding recombinant nucleic acids, due to the possibility of horizontal gene transfer (e.g., antibiotic resistance genes, especially as mediated by plasmids)[57,58]. Our kill switch design mitigates this possibility in that the only plasmid-borne elements include the Tet repressor, which does not alone confer antibiotic resistance, and the gRNAs, which carry inherent specificity to the EcN genome and whose transcription should have no cleaving effect in the absence of the Cas9 cassette. As such, the incorporation of the *infA* plasmid-maintenance system served to both improve stability of the kill switch and reduce the risk of antibiotic resistance gene dis-semination. We included plasmid-borne spectinomycin and chloramphenicol resistance genes to selectively distinguish our control and CRISPRks strains in co-culture and co-gavage experiments; clinical iterations of the kill switch strains for practical applications would omit these selective markers.

Inspired by the competitive inter-strain exclusion properties observed in gut pathogens[59] and commensals alike[31,60], we hypothesized genetic approaches for mitigating kill switch escape could be complemented at the population level by external competitive pressure. With the goal of eliminating a specific subpopulation of engineered microbes and not a probiotic species itself from the gut, we provided both CRISPRks EcN and control EcN in our animal models, and demonstrated that kill switch

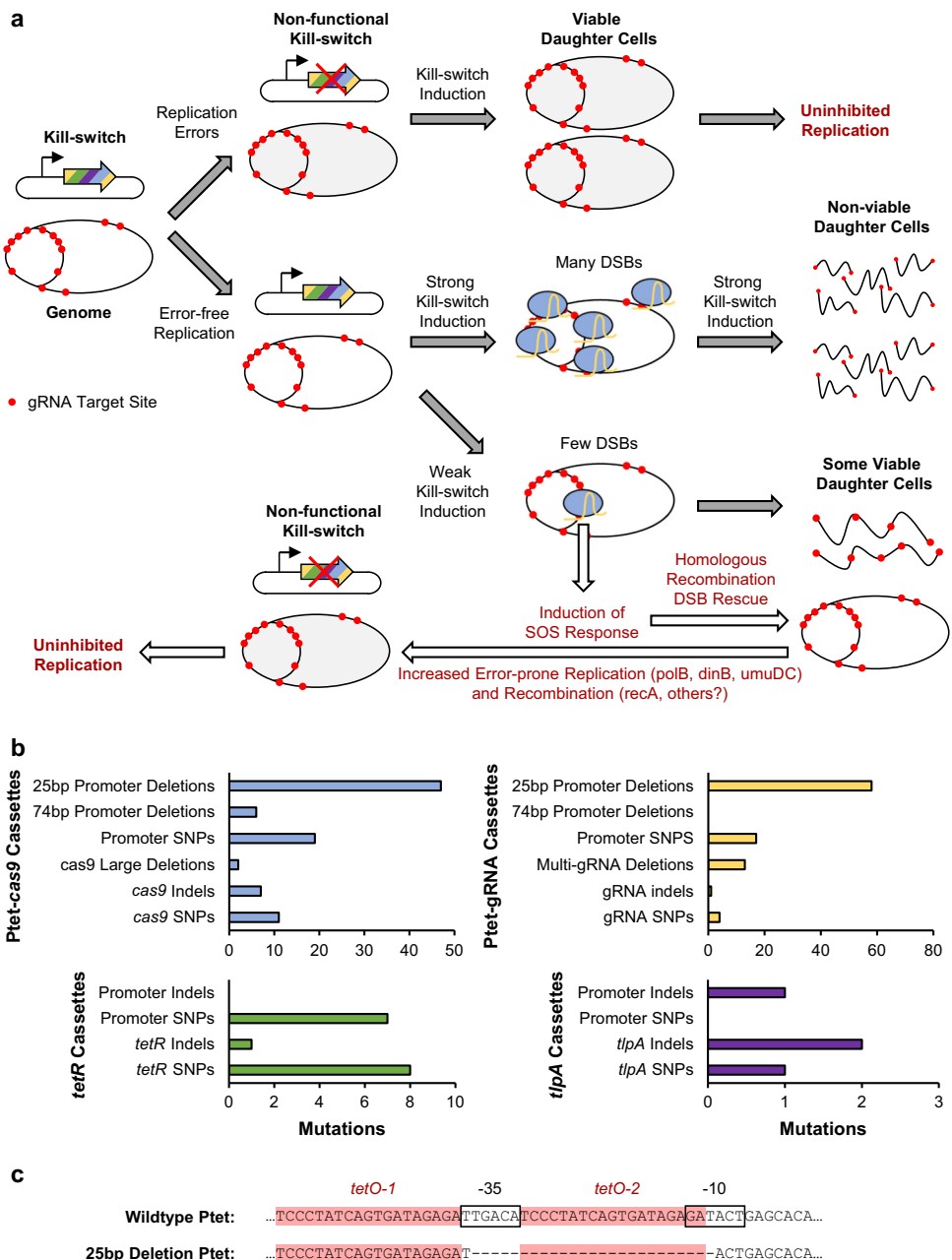

**Fig. 6 Summary of sequencing results from non-functional kill switches and potential mechanisms of circuit inactivation. a** Potential mechanisms of kill switch inactivation and cell survival. DSB, double-strand break. **b** Classifications of the observed mutations in $P_{tet}$-*cas9*, $P_{tet}$-gRNA, *tetR*, and *tlpA* expression cassettes. Source data are provided as a Source Data file. **c** Sequence of the most commonly observed inactivation mutation in the utilized $P_{tet}$ promoters.

induction in synergy with competitive exclusion enabled virtually complete eradication of the kill switch population. Co- or pre-administration of a wild-type probiotic strain in clinical applications of engineered probiotics could help prime the gut for induced elimination of the engineered probiotic. However, ways to ensure the stability of both populations prior to kill switch induction must be further investigated.

The performance of the kill switches could be further improved by addition of an orthogonal kill switch mechanism, an approach that has recently been demonstrated to have a multiplicative effect on efficacy[18]. In the absence of aTc exposure in the gut (i.e., induction only with temperature upon excretion), there was an increasing incidence over time of biocontainment failure for the 2-input CRISPRks (Fig. 5b, bottom), suggesting that the strain requires both induction modalities to perform robustly. Indeed,

the 2-input CRISPRks also did not achieve full efficacy in vitro when induced with temperature alone (Fig. 4b), in contrast to the single-input kill switch when induced with aTc in vitro (Fig. 2c), indicating that $P_{tlpA}$-driven expression of tetR may be leaky even at low temperatures or that the TetR degradation rate is not sufficiently high to robustly induce kill switch activation. Because the temperature- and aTc-driven induction feed into the same kill switch mechanism, the combination of both inputs does not result in a synergistic increase in efficacy. Similarly, induction with both temperature and aTc in the context of intra-niche competition led to no detectable growth after direct plating of feces (approximately 2% of the entire fecal sample at the 1X dilution), nor after inoculating 24 h growth assays at room temperature to allow any putative escape mutants to amplify; the exception was at the 72 h timepoint, where after 24 h of growth in

rich media, we observed a mean 583 CFU/mL of the 2-input CRISPRks (Fig. 5f). With a room temperature doubling time of 115 min estimated from control strain titers from the same mice at the same timepoint, this implies a baseline CRISPRks titer of 0.1 CFU/mL, or approximately 100 CFU in the entire fecal sample given our inoculation scheme. This was not captured when plating directly from feces (Fig. 5b, bottom), highlighting the importance of survival assay sensitivity in the development of kill switches. Escape events such as this could also be mitigated by addition of an orthogonal kill switch circuit, in addition to further optimization of the *tlpA* and *tetR* cassettes.

In summary, we developed CRISPR-based kill switches in EcN to create a safe probiotic chassis for future biomedical technologies. These kill switches allow EcN to proliferate under normal gut conditions and initiate cell death in response to oral consumption of an inducer and excretion from the body. We have demonstrated a kill switch approach to on-demand selective removal of engineered microbes from the gut. We also explored diverse methods for improving the stability of the kill switches and minimized key mechanisms of kill switch inactivation. The engineered kill switches apply genetic parts (CRISPR/Cas9 and TetR/Ptet) that have been shown to be functional in diverse microbes[32–34]. As such, similar kill switches can be engineered for a larger panel of probiotic microbes. Furthermore, the temperature sensing module can be replaced with sensors for alternative environmental conditions[46] and chemicals[61–63] to create 2-input kill switches for novel applications. These microbial biocontainment tools will facilitate the creation of living therapeutics or other microbes for environmental applications that are more robust, predictable, and controllable, which is critical for both regulatory approval and public acceptance of genetically modified organisms.

## Methods

**Experimental model and subject details**. All in vitro experiments were performed in compliance with the Washington University in Saint Louis Institutional Biological & Chemical (IBC) Safety Committee. All plasmids were assembled in and purified from *E. coli* DH10B. Purified plasmids were subsequently transferred to and tested in wild-type or engineered *E. coli* Nissle 1917 variants lacking the two native plasmids, pMUT1 and pMUT2.

All mouse experiments were approved by the Washington University in Saint Louis School of Medicine Institutional Animal Care and Use Committee (Protocol number: 21-0160), and performed in AAALAC-accredited facilities in accordance with the National Institutes of Health guide for the case and use of laboratory animals.

All mouse experiments were performed in female 8-week old C57BL/6 mice (Jackson Labs C57BL/6 J, RRID:IMSR_JAX:000664). Mice were housed in a specific pathogen free barrier facility maintained by WUSM DCM at 30-70% humidity and 68–79 °F under a 12:12 hour light:dark cycle. Mice were provided feed (Purina Conventional Mouse Diet (JL Rat/Mouse 6 F Auto) #5K67) and water *ad libitum*. Mice were co-housed with up to 5 mice per cage, and at least 2 cages per experimental arm to account for cage effects.

Oral gavage of mice was performed using 18ga x 38 mm plastic feeding tubes (FTP-18-38, Instech). To ablate the native microbiome prior to EcN colonization, each mouse was administered 20 mg streptomycin sulfate salt (S6501, Sigma-Aldrich) in 100 µL H2O via oral gavage[44]. Streptomycin treatment of mice, often used in *Salmonella enterica* colonization models[44], has also been used extensively in colonization models for commensal and pathogenic strains of *Escherichia coli*[13,31,64–68]. 24 or 72 hours after streptomycin administration, mice were orally gavaged with $10^8$ CFU EcN in 100 µL phosphate buffered saline (PBS). To test alternative microbiome ablation strategies, mice were instead orally gavaged with 6 mg carbenicillin disodium salt (C1389, Sigma-Aldrich) in 100 µL H2O 48 hours prior to EcN administration[45], or left untreated prior to EcN administration.

24 hours after EcN administration, mice were switched from standard drinking water to aTc treatment water ($10^5$ ng/mL aTc + 5% sucrose, filter sterilized) or control water (5% sucrose, filter sterilized), provided *ad libitum*. Treatment water and control water were prepared fresh and replaced daily for the duration of each experiment. Fecal samples were collected at the indicated timepoints. At the end of each experiment, mice were sacrificed through carbon dioxide asphyxiation.

## Method details

*Plasmids, strains, and growth conditions.* All plasmids were designed using Snap-Gene and assembled in *E. coli* DH10B using the Gibson Assembly (100 mM Tris-

HCl, 10 mM MgCl2, 0.2 mM dNTPs, 10 mM DTT, 5% PEG-8000, 1 mM NAD+, 4 U/µL Taq DNA ligase, 4 U/mL T5 exonuclease, 25 U/mL Phusion DNA polymerase) or Golden Gate Assembly (1X T4 ligase buffer, 1X Cutsmart buffer, 40 U/µL T4 ligase, 1 U/µL SapI, 1 U/µL DpnI) methods. Wild-type or engineered EcN variants were then transformed with the purified and sequence-verified plasmids for kill switch testing. Plasmid DNA was isolated using the PureLink Quick Plasmid Miniprep Kit (K210011, Invitrogen), and polymerase chain reaction (PCR) products were extracted from electrophoresis gels using the Zymoclean Gel DNA Recovery Kit (D4008, ZYMO research). Enzymes were purchased from New England Biolabs (Ipswich, MA, USA).

To construct engineered EcN variants, we utilized lambda red-mediated CRISPR-Cas9 recombineering as previously described[69]. To create knockouts of specific genes (Supplementary Table 2), gRNAs for the pgRNA plasmid were designed using the gRNA designer from Atum (atum.bio) to target the gene of interest. 60 bp ssDNA oligos were designed with 30 bp arms homologous to the lagging strand of DNA synthesis flanking the region to be knocked out. For insertions (Supplementary Table 3), the pgRNA plasmid was similarly constructed. The dsDNA insert was obtained by constructing a plasmid with the DNA to be inserted flanked by 500 bp arms homologous to the insertion region. The full product (both arms and insert DNA) were PCR amplified and purified by gel extraction. EcN harboring pMP11 was then transformed with 100 ng of the respective pgRNA plasmid and either 1 µM of the ssDNA oligo or 100 ng of the dsDNA insert to initiate recombination. All sequencing (Supplementary Table 4) was performed by Genewiz (South Plainfield, NJ, USA). Primers were purchased from Integrated DNA Technologies (Coralville, IA, USA). All plasmids and parts constructed and used in this work are summarized in Supplementary Tables 5, 6, respectively.

Unless otherwise specified, LB medium was used for culturing. For Figs. 2f, 3a, Supplementary Fig. 4a, c, M9 minimal medium supplemented with 1 mM MgSO4, 100 µM CaCl2, and 0.4% w/v glucose was used. 0.2% w/v casamino acids was also added to the M9 minimal medium as specified. Medium was supplemented with the following concentrations of antibiotics as necessary: 100 µg/mL ampicillin, 34 µg/mL chloramphenicol, 20 µg/mL kanamycin, 10 µg/mL gentamycin, and 100 µg/mL spectinomycin (Gold Biotechnology, Olivette, MO, USA). Unless otherwise stated, cultures were incubated at 37 °C with 250 rpm shaking.

*Standard kill switch assays.* EcN was transformed with kill switch plasmids by electroporation and plated on LB agar with the relevant antibiotics. Single colonies were then transferred to 1 mL of LB in 14 mL round bottom tubes (14-959-11B, Fisher Scientific) and grown in a shaking incubator at 37 °C and 250 rpm for ~2 h until exponential phase (OD600 of 0.25-0.50) was reached. To test for aTc-inducible cell death, these cultures were diluted to an OD600 of 0.01 in 1 mL fresh LB medium with the specified concentrations of aTc[28]. At the indicated timepoints, samples were removed from the cultures for viable colony forming unit (CFU) quantification. CFUs were determined by plating 10 µL of serially diluted cultures onto LB agar with the relevant antibiotics unless otherwise specified and incubating at 37 °C overnight. For cultures where no colonies were obtained in the undiluted sample, 100 µL was also plated to ensure a more accurate quantification. The fraction of viable cells was calculated as the ratio of CFUs obtained from the induced culture to the number of CFUs obtained from the uninduced culture. To test for temperature-inducible cell death, the exponential phase cultures were serially diluted and plated onto two LB agar plates. One plate was incubated at 37 °C overnight while the other plate was incubated at 30 °C for two weeks. For temperature-sensitive assays, all liquid medium and LB agar plates were preheated to 37 °C for at least two hours prior to the addition of cells.

*Long-term stability kill switch assay.* A long-term stability assay was used for Figs. 2e, 4d, Supplementary Figs. 3e, 6e. On day 1, single colonies were transferred to 1 mL LB without antibiotics, incubated until an OD600 of 0.25–0.50 was reached, and diluted to an OD600 of 0.01 for the standard aTc kill switch assay or directly plated onto LB agar for the standard temperature kill switch assay as before. The original undiluted cultures were then returned to the shaking incubator and cultured for an additional ~22 h (24 h of incubation total). After the 24 h incubation, each culture was diluted 250X into fresh LB without antibiotics and incubated for an additional 24 h. These daily dilutions were repeated for a total of 28 days. Every 3–4 days, the culture was used for the kill switch assays as described above. On assay days, culture samples were stored at −80 °C in 15% (v/v) glycerol.

*Temperature-sensing transfer curve kill switch assay.* For temperature-sensitive transfer curves, single colonies were then transferred to 1 mL of LB in 14 mL round bottom tubes and grown in a shaking incubator at 37 °C and 250 rpm for ~2 h until exponential phase (OD600 of 0.25-0.50) was reached. These cultures were then diluted to an OD600 of 0.01 in 50 µL of fresh LB medium in PCR tubes and incubated at the specified temperatures in a thermocycler for 5 h. CFUs were determined by plating 10 µL of serially diluted cultures onto LB agar and incubating at 37 °C overnight.

*In vitro condition-poor competition kill switch assay.* EcN strains with and without the Δrpdu knockouts were transformed with the no gRNA control and the aTc-inducible CRISPRks. Single colonies for each of the four strains were grown overnight

in 5 mL LB at 37 °C and 250 rpm. The following day, each culture was centrifuged at 3,000 g, the LB supernatant was removed, and the cell pellet was resuspended in 5 mL M9 + 0.4% glucose. For CRISPRks-only cultures, 1.5 mL of the respective CRISPRks strain was diluted into two 15 mL conical tube at a final volume of 15 mL M9 + 0.4% glucose. For competitive mixture cultures, 0.75 mL of each CRISPRks strain was also diluted at a 1:1 ratio with the respective no gRNA control strain (1.5 mL total culture) into two additional 15 mL conical tubes. One of the two conical tubes in each pair was induced with 500 ng/mL aTc. Each tube was capped to stop oxygen transfer and incubated at 37 °C without shaking. Every 24 h for 72 h, the cultures were removed from the incubator, the CFUs were quantified, and the cells were diluted 10X into a new tube with fresh M9 + 0.4% glucose and 500 ng/mL aTc.

*Assays of intestinal samples for quantification of viable EcN.* Fecal samples (and cecal samples post-sacrifice) were collected in pre-weighed sterile 2 mL microtubes, and weighed again to accurately measure sample mass. Samples were homogenized in 500 μL PBS on a benchtop vortexer (2000 rpm for 3 min), serially diluted in PBS, and plated on LB agar supplemented with 100 μg/mL spectinomycin dihydrochloride pentahydrate (S4014, Sigma-Aldrich) to select for spectinomycin-resistant kill switch strains, and 34 μg/mL chloramphenicol (AC227920250, Acros Organics) to select for chloramphenicol-resistant control strains. All samples were plated on both chloramphenicol- and spectinomycin-supplemented LB agar plates to confirm no cross-contamination of strains between experimental arms. Dilutions of fecal or cecal homogenates plated for all experiments spanned $10^0$ to $10^{-6}$ (i.e. undiluted homogenate was plated for each sample). CFU/mg sample values were calculated from enumerated colonies. Two technical replicates per intestinal sample were processed.

For experiments with the single-input CRISPRks (aTc-inducible only) strains, samples were weighed, resuspended, serially diluted, and plated at room temperature (22 °C), and plates were incubated overnight at 37 °C prior to colony enumeration. For experiments with the 2-input CRISPRks (aTc- and temperature-inducible) strains, fecal samples were immediately placed in a pre-warmed heatblock (37 °C) inside an insulated container. Similarly, to maintain cecal samples at 37 °C, freshly sacrificed mice were kept in a 37 °C incubator until dissection and collection of cecal contents. Samples were weighed, resuspended, serially diluted, and plated inside a 37 °C warm room, using pre-warmed PBS and LB agar plates. After plating, LB plates continued to be incubated at 37 °C overnight prior to colony enumeration. To assay temperature-dependent induction of the kill switch, samples were then moved to room temperature (22 °C) and again plated on spectinomycin- or chloramphenicol-supplemented LB agar plates. Plates were incubated at 22 °C for 48 h prior to colony enumeration.

To assay biocontainment efficacy of the 2-input CRISPRks strains via room temperature kinetic growth assays, 10 μL of the 100X dilution of each sample was inoculated into 190 uL LB broth supplemented with 100 μg/mL spectinomycin or 34 μg/mL chloramphenicol in a 96-well plate, and read in a plate reader (Biotek Powerwave HT) at room temperature (22 °C) for 24 h (kinetic read, 5 sec shake followed by absorbance reading at 600 nm [Abs600] at 20 min intervals). Additionally, at 3 h and 24 h, kinetic growth assay cultures were sampled, serially diluted, and plated on LB agar supplemented with 100 μg/mL spectinomycin or 34 μg/mL chloramphenicol, and incubated overnight at 37 °C prior to colony enumeration.

*Kill switch functionality assay.* To assess whether colonies that survived kill switch assays maintained functional kill switches, three no gRNA control colonies and the specified number of surviving kill switch colonies were transferred to 600 μL LB in 96-deep well plates (E951032808, Fisher Scientific). The plates were then incubated at 37 °C and 250 rpm for 3 h and diluted 20X into 600 μL of fresh LB with and without 500 ng/mL aTc. After 3 h of induction, the Abs600 was measured for each culture, and the ratio of uninduced Abs600 to induced Abs600 was determined. Colonies with non-functional kill switches were defined as having Abs600 ratios within three standard deviations of the Abs600 ratio of the no gRNA control.

*Transforming gRNA plasmids into Ptet-cas9 integration strains.* A temperature-curable TetR expression plasmid was constructed with a constitutive TetR expression cassette, a temperature-sensitive oriR101 origin, and kanamycin resistance gene (pAGR377). EcN strains with Ptet-cas9 genomic integrations were transformed with this plasmid, plated on LB agar with kanamycin, and incubated overnight at 30 °C. Cells with the pAGR377 were then made electrocompetent and transformed with the gRNA kill switch plasmids. Transformed cells were recovered in 600 μL SOC at 30 °C for 1 h. The cells were then plated onto LB agar with spectinomycin to select for the kill switch plasmids and incubated overnight at 42 °C to cure pAGR377. Successful curing of pAGR377 was confirmed by streaking colonies on agar plates with spectinomycin only and with both spectinomycin and kanamycin.

*Generating antibiotic-independent kill switches.* To remove the antibiotic dependence of the kill switches, the *infA* essential gene was knocked out of the genome and constitutively expressed on the plasmid of interest[39,40]. To generate an *infA* knockout EcN strain, the pMP11 CRISPR plasmid first was modified to constitutively express InfA (pAGR309). The recombination protocol described above was then used to perform and confirm the knockout. Next, the TetR expression plasmid was modified to also constitutively express InfA (pAGR384). The *infA* knockout strain containing the modified pMP11 plasmid was then transformed with AGR384 and plated on LB agar with kanamycin. The transformants were

incubated overnight at 37 °C to allow for moderate curing of the ampicillin-resistant modified pMP11 plasmid. Colonies from the plate were streaked onto LB agar plates with and without ampicillin to identify colonies with successful curing of the modified pMP11 plasmid. gRNA kill switch plasmids were constructed containing another constitutive InfA expression cassette. Cells containing only pAGR384 were transformed with the InfA-expressing gRNA kill switch plasmids and incubated overnight at 42 °C to cure pAGR384. Successful curing of pAGR384 was confirmed by streaking colonies on agar plates with spectinomycin only and with both spectinomycin and kanamycin.

*Growth and fluorescence measurements.* Population absorbances at 600 nm (Abs600) were measured in 96-well black assay microplates (07-000-088, Fisher Scientific) using a Tecan microplate reader (Infinite M200 Pro) and Tecan i-Control and converted to optical density OD600 when necessary. To measure GFP fluorescence, culture samples were transferred to 200 μL filtered 0.9% (w/v) saline supplemented with 2 mg/ml kanamycin in 96-well clear round bottom assay microplates (353910, Corning) for measurements. Flow cytometry analysis was carried out using a Millipore Guava EasyCyte High Throughput Flow Cytometer and Guavasoft 2.7 software with a 488 nm excitation laser and a 512/18 nm emission filter. 10000 events for each sample, gated by forward (minimum/maximum of 20/600) and side scatter (minimum/maximum of 30/2000), were measured at a flow rate 0.59 μl/s. FlowJo (TreeStar Inc.) was used to obtain the average fluorescence of the population. The fluorescence (au) of each sample was calculated using the following formula: $F_s = F_{experiment} − F_{EcN}$, where $F_s$, $F_{experiment}$, and $F_{EcN}$ respectively represent the reported sample fluorescence, measured sample fluorescence, and autofluorescence (background fluorescence of EcN lacking GFP).

*Characterization of TlpA temperature sensors in EcN.* The $P_{tlpA}$ promoter and *tlpA* gene were PCR amplified from *Salmonella typhimurium* SL1344 genomic DNA and inserted upstream of *gfpmut3* to form a negative feedback operon. To convert *tlpA* to the modified *tlpA* (translated to the protein TlpA*) developed by Pirner et al., the following amino acid substitutions were inserted into the *tlpA* gene: P60L, G135V, K187R, K202I, and L208Q[30]. After transforming EcN with each plasmid, single colonies were transferred to 5 mL of LB and incubated overnight at 37 °C and 250 rpm. The following day, cultures were diluted 1000X into 50 μL of LB in PCR tubes[30] (T3202N, Fisher Scientific). The PCR tubes were incubated in a thermocycler running temperature gradients between 31.5–45.5 °C (TlpA) or 31–37.5 °C (TlpA*) for 24 h. The GFP fluorescence of the cultures was then quantified by flow cytometry as described above.

*Incorporating and optimizing temperature sensing in the kill switch.* The $P_{tlpA}$-*tlpA** negative feedback cassette was inserted in place of the constitutive promoter controlling TetR expression on the gRNA kill switch plasmid. To optimize the expression and stability of TetR in the kill switch, a *tetR* RBS library and a C-terminal SsrA degradation tag library were simultaneously inserted into the plasmid (Supplementary Table 6). The RBS library was designed using the De Novo RBS Library Calculator v2.1 to have 128 different variants with translation initiation rates spanning 20−100,000 au[70]. The initial RBS had a translation initiation rate of 1,500 au. The SsrA degradation tag library was constructed by randomly mutagenizing the third to last codon of the SsrA degradation tag[71]. The total number of potential unique construct variants was 3,328. The library was generated in *E. coli* DH10B and tested in EcN.

*Hill equation fitting.* The Hill equation (Eq. 1) was used to fit lines to the fluorescence, CFU, and fraction viable data. The model was fit to the experimentally collected data by minimizing the root mean square error (RMSE; Eq. 2). Fitted values are listed in Supplementary Table 1.

$$F = F_{min} + \frac{(F_{max} - F_{min})}{1 + \left(\frac{K_A}{[L]}\right)^n} \qquad (1)$$

where

$F$ = Calculated fluorescence, CFUs/mL. or Fraction Viable
$F_{max}$ = Maximum fluorescence, CFUs/mL. or Fraction Viable
$F_{min}$ = Minimum fluorescence, CFUs/mL. or Fraction Viable
$K_A$ = Half maximal constant
$n$ = Hill coefficient
$[L]$ = Ligand concentration or temperature

$$RMSE = \sqrt{\frac{\sum_{N=1}^{N}\left(F - F_{exp}\right)^2}{N}} \qquad (2)$$

where

$RMSE$ = Root mean squared error
$F$ = Calculated fluorescence, CFUs/mL, or Fraction Viable
$F_{exp}$ = Actual experimental fluorescence, CFUs/mL, or Fraction Viable
$N$ = Number of data points

**Quantification and statistical analysis.** All statistical tests were performed using GraphPad Prism or Excel. All statistical details of experiments, including significance criteria, sample size, definition of center, and dispersion measures can be found in the figure legends, in the Results section, or in the Source Data file. Sample sizes for animal experiments, reporter assays, and viability assays were chosen based on our previous work[13,72] and the literature, and represent sample sizes routinely used for these methods. No sample size calculations were performed during the design of experiments. Samples were randomized during group assignment in all experiments. No samples were excluded from analyses. The Investigators were not blinded to allocation during experiments and outcome assessment.

**Reporting summary.** Further information on research design is available in the Nature Research Reporting Summary linked to this article.

## Data availability

All source data and plasmid maps have been deposited to Mendeley Data (DOI: 10.17632/dwhfp2ycyw.1). Any additional information is available from the Lead Contact upon request. Plasmids and strains generated in this paper are available upon request from the Lead Contact. This study did not generate additional new unique reagents. Source data is available in the Source Data file. Source data are provided with this paper.

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

## Acknowledgements

We thank Prof. Brian Pfleger for the gift of the pMP11 and pgRNAcm plasmids. We also thank Juya Jeon for her assistance in assembling the temperature-sensing GFP reporter plasmids. We thank members of the Moon and Dantas labs for helpful suggestions and comments on this work and manuscript. This work was supported by the National Institutes of Health (R01 AT009741 to T.S.M. and G.D.), the Office of Naval Research (N00014-17-1-2611 and N00014-19-1-2357 to T.S.M.), the United States Department of Agriculture (2020-33522-32319 to T.S.M.), National Science Foundation (CBET-1350498 to T.S.M.) and U.S. Environmental Protection Agency (84020501 to T.S.M.). The content is solely the responsibility of the authors and does not necessarily represent the official views of the funding agencies.

## Author contributions

T.S.M. conceived the project. A.G.R., A.F., S.F., G.D. and T.S.M. designed experiments and analyzed the data. A.G.R., A.F and S.F. performed the experiments. A.G.R., A.F., S.F., G.D. and T.S.M. wrote the manuscript.

## Competing interests

The authors declare no competing interests.
