## [Peer Review File · Nature Communications]

Reviewers' Comments:

Reviewer #1:

Remarks to the Author:

In this work, the authors strive to create a kill switch for microbial biological containment that would function stably in a complex environment, namely the mouse gut. They report numerous designs and iterations in a highly logical and thorough fashion, focusing primarily on the use of CRISPR to mediate a single-input kill switch and secondarily on a temperature-responsive switch in *E. coli* Nissle 1917. Their early design and characterization sheds light on the type and frequency of mutations that can break the system, forcing the authors to include functional redundancy. After noting that under non-permissive conditions mutation rates can rise, they also delete several genes known to mediate the SOS response. While these changes as well as promoter modifications and antibiotic-independent plasmid retention lead to impressive kill switch performance in aerobic bacterial cultures, the authors next pursue in vitro models that are closer to mimicking the gut and find that their system fails. It is here that they begin to employ the strategy of intra-niche competition using the wild-type Nissle strain (or slightly modified equivalents). This strategy is mostly effective and reveals an important lesson about how escape mutants can be prevented from taking over a population; however, it is an acknowledgement that their intrinsic biological containment system is unable to do the job alone. This is acceptable because, while certain synthetic biology strategies probably could accomplish this feat without need for the intra-niche competition, the intra-niche competition is a viable strategy for delivery of real probiotics, and the strides reported in kill switch technology development in this manuscript are very impressive. Ultimately, the authors find that strains harboring either kill switch are effectively and selectively eliminated inside the murine gut, whereas strains harboring the 2-input switch are also eliminated during excretion (although there are occasional exceptions to this, such as the escape mutants observed at 72 hours in Figure 5F).

The work tackles a grand challenge in synthetic biology of engineering stable systems to regulate microbial proliferation in settings pertinent to environmental or human health. The experiments conducted are well organized and support all the claims made in the paper. The figures – both main text and SI – are generally exceptionally well done. The writing quality is high. And ultimately, though some aspects of the biological containment problem are not completely resolved, the body of work and the final kill switch performances are high impact contributions to the field. As such, this reviewer has only minor comments that will help audiences better understand the results and the broader implications of this work.

- The first minor comment is that this seems like an unusual circuit design for a kill switch for biological containment. Typically, the input for a kill switch is something that is present in a specific environment where the organism is intended to grow. Usually the lethal components are expressed when the organism leaves that environment. Such systems try to provide real biological containment in that they prevent proliferation in the wild. What the authors report here is a strain that they've made sensitive to anTc, almost as if it were an organism-specific antibiotic. Conceptually, this is interesting because it means a microbe could proliferate in the gut for potentially a long time, performing its actions before a patient chooses to end its survival. It is complementary to many existing kill switches. But practically, such a system has a major limitation that the organism could proliferate in any environment upon excretion (i.e. the default condition is permissive because anTc would be absent). That makes the 2nd input of temperature-response a very critical component in ensuring that genetically modified microbes are not released in the wild after excretion. The authors should further discuss the rationale for their design and inputs more in the introduction (pros and cons), both in reference to traditional kill switches and in motivating why they looked at the 2nd input, because it currently seems like an afterthought.
- The literature survey of biocontainment systems is good but misses a few more recent references, particularly in regard to where stability was investigated. The Silver group has reported more recent kill switches (Stirling et al 2020). Similarly, the Church group has reported stability evaluation of auxotrophy based on synthetic amino acids (Kunjapur et al 2021). Also, the introductory text suggests that all auxotrophy can be overcome by cross-feeding but this limitation does not apply when synthetic amino acids are used.
- There were a few figures that were difficult to understand with the level of explanation currently provided. Figure S1D was one such example. Please elaborate there.

- More information should be provided in relation to the antibiotic-independent *infA* construct. It is currently brought up in Line 204, with ambiguity about the cloning site and strength of the constitutive promoter (it is possible that I missed this). Additionally, it looks like antibiotic was still added in some experiments that used this construct (Figs S2B, S2C, S3C), so does the construct still contain an antibiotic resistance marker?
- aTc concentrations vary in Fig 1C/1H, which is alluded to in the Fig 1 caption. Could the authors comment on the reason for this change – was it simple assay optimization or something else?
- The specific gRNA targets that are used in various assays could use more clarity. In Fig 1C, *groL-2*, *ile-2*, and *rrs-2* all seem comparable as targets. It is clear from the text and from the figure that *groL-2* and *ile-2* are pursued further as targets first (e.g. Fig 1H). Then, the text seems to make it clear that *rrs-2* is tested to improve the *infA* construct; but neither Fig 2 or S2 seem to specify the gRNA targets (they say X3 instead of X2, but in all other places this refers to redundancy, whereas these are distinct targets). After Fig 2, it is unclear if all 3 gRNA targets are consistently used in the subsequent kill switches. It is also unclear what the Mut14 variant is (what has been mutated, to what sequence?).
- This is not a request for the authors to perform additional experiments, however this reviewer is very curious whether other ratios of co-delivered strains were tested besides 1:1, even in the in vitro assay? Given how effective and important this approach is, it would be nice to know how tuning of the ratio influences escape.
- In Fig 3G, the 24 hour (del)rpdu error bar is massive, and at other time points the performance is equivalent to the other strain. Can any meaningful conclusions be drawn from this data?
- In Fig 3J, why is there a lack of CFUs/mg in the control settings? Is this an indication of a lack of colonization by a codelivered strain? This seemed unexpected but it is possible that I missed something.
- 5B: The OR behavior (atC - , 2-input CRISPRs) is visibly weaker than the AND gate expected upon excretion, which is fair. However, is this good enough for biocontainment of an engineered probiotic? It seems like it will proliferate upon excretion until aTc is added.
- 5F: As mentioned in the text, the 2-input kill switch leads to escape under one of the conditions measured – the 72 h timepoint. The discussion of this result seems limited. What are the implications for how this switch could be applied, and what next steps if any might be needed for improvement? Since intra-niche competition is already being used, could the authors offer a brief recommendation on what else should be considered by future researchers to prevent escape?

Reviewer #2:

Remarks to the Author:

In this manuscript, the authors reported two “CRISPR-based kill switches” in *E. coli* Nissle 1917 (EcN), that could selectively terminate the host cells, upon anhydrotetracycline induction or exposure to low temperature (<33°C). To ensure effective killing by the kill switches, the authors performed extensive optimization on the genetic circuits and the chassis strain, including development of antibiotic-independent plasmid system, implementation of functional redundancy with the Cas9 expression cassette, modulation of bacterial SOS response and provision of intra-niche competition by a closely related strain. The optimized strains with embedded kill switches were then tested in the mice model to show efficient elimination of EcN, upon induction. Overall, the manuscript is well-written. The findings of the study are well-presented, and the conclusions drawn by the authors are well-founded. Please find below my comments.

1. Would the implementation of kill switches potentially impair the capability of therapeutics production in EcN? The authors have edited EcN genome to a great extent to ensure the efficiency of the kill switches, which included integration of the Cas9 expression cassette, deletion of essential gene *infA* and deletion of SOS response genes. Considering that these modifications can affect essential cellular functions, have the authors considered the impact of these modifications on the therapeutic production capability of EcN?
2. As the kill switches are intended to serve as a biocontainment mechanism for engineered therapeutic EcN strains, I suggest the authors to characterize the impact of the kill switches on protein productions in EcN. This can be done by either comparing the expression level of native proteins or co-expression of reporter proteins in the engineered EcN strain.

3. One of the major concerns in the applications of genetically modified microorganisms is a potential horizontal gene transfer that can occur amongst engineered strains to surrounding native microbiome. Such a phenomenon has been previously reported in two separate studies (<https://pubmed.ncbi.nlm.nih.gov/17014680/>; <https://pubmed.ncbi.nlm.nih.gov/18238887/>). In the case of kill switches, would the genetic materials, especially plasmids, be released to the native microbiome following bacterial cell death? If so, will it increase the chances of horizontal gene transfer? Authors should discuss such possibilities and elaborate on how to minimize horizontal gene transfer event that can occur with these kill switches.

4. The authors have engineered a number of EcN strains with various genetic circuits. To make it easier for the readers to follow, the authors can include the corresponding strain names in the schematics of the genetic circuits in the figures (eg. in Figure 1A, Figure 2A B C, and Figure 4A).

5. Line 257-259 "Thus, we hypothesized that a nutrient- and growth-limited environment impairs the DNA-cleavage rate of the kill switch and allows the survival of daughter cells with an induced SOS response and elevated kill switch inactivation rate." Further explanation is needed to understand this hypothesis. Please explain why the growth-limited environment impairs the DNA-cleavage rate and how this is related to SOS response genes. Secondly, if the DNA-cleavage rate was indeed impaired, shouldn't it lead to a weaker or slower SOS response?

6. Line 364 -366, "We tested over 500 2-input CRISPRs Δ rpdu variants". A few different methods were applied to improve the temperature response in the engineered EcN strain. Please provide more detailed and descriptive term than "CRISPRs Δ rpdu variants" to provide better understanding of the range of EcN strain variants being tested?

7. Line 802 - 804, streptomycin was administered to mice to enable EcN colonization. The reference cited, however, used the same treatment to enable the colonization of a *Salmonella enterica*. Please provide more relevant citation (i.e. for EcN colonization).

Reviewers of manuscript NCOMMS-21-35957-T
November 9, 2021

Dear Reviewers & Ross,

We thank you for your insightful comments and critiques, and the time taken to review our submission. In response to your suggestions, we have reworked the introduction to better highlight the field of biocontainment and the uniqueness of the circuits described here, provided more clarity and detail on the experiments performed and results obtained, further investigated the capabilities of the described co-administration technique and the protein production capacity of the engineered strains, and provided additional discussion of the hypotheses explored and the potential shortcomings of the kill switches. Below, we address each of your major and minor comments specifically. With more data and discussion, we believe this manuscript is very strong. Thank you again for your helpful comments.

Reviewer #1 (Remarks to the Author):

In this work, the authors strive to create a kill switch for microbial biological containment that would function stably in a complex environment, namely the mouse gut. They report numerous designs and iterations in a highly logical and thorough fashion, focusing primarily on the use of CRISPR to mediate a single-input kill switch and secondarily on a temperature-responsive switch in *E. coli* Nissle 1917. Their early design and characterization sheds light on the type and frequency of mutations that can break the system, forcing the authors to include functional redundancy. After noting that under non-permissive conditions mutation rates can rise, they also delete several genes known to mediate the SOS response. While these changes as well as promoter modifications and antibiotic-independent plasmid retention lead to impressive kill switch performance in aerobic bacterial cultures, the authors next pursue *in vitro* models that are closer to the mimicking the gut and find that their system fails. It is here that they begin to employ the strategy of intra-niche competition using the wild-type Nissle strain (or slightly modified equivalents). This strategy is mostly effective and reveals an important lesson about how escape mutants can be prevented from taking over a population; however, it is an acknowledgement that their intrinsic biological containment system is unable to do the job alone. This is acceptable because, while certain synthetic biology strategies probably could accomplish this feat without need for the intra-niche competition, the intra-niche competition is a viable strategy for delivery of real probiotics, and the strides reported in kill switch technology development in this manuscript are very impressive. Ultimately, the authors find that strains harboring either kill switch are effectively and selectively eliminated inside the murine gut, whereas strains harboring the 2-input switch are also eliminated during excretion (although there are occasional exceptions to this, such as the escape mutants observed at 72 hours in Figure 5F).

The work tackles a grand challenge in synthetic biology of engineering stable systems to regulate microbial proliferation in settings pertinent to environmental or human health. The experiments conducted are well organized and support all the claims made in the paper. The figures – both main text and SI – are generally exceptionally well done. The writing quality is high. And ultimately, though some aspects of the biological containment problem are not completely resolved, the body of work and the final kill switch performances are high impact contributions to the field. As such, this reviewer has only minor comments that will help audiences better understand the results and the broader implications of this work.

We thank the reviewer for this very positive and detailed comment.

1. The first minor comment is that this seems like an unusual circuit design for a kill switch for biological containment. Typically, the input for a kill switch is something that is present in a specific environment where the organism is intended to grow. Usually the lethal components are expressed when the organism leaves that environment. Such systems try to provide real biological containment in that they prevent proliferation in the wild. What the authors report here is a strain that they've made sensitive to aTc, almost as if it were an organism-specific antibiotic. Conceptually, this is interesting because it means a microbe could proliferate in the gut for potentially a long time, performing its actions before a patient chooses to end its survival. It is complementary to many existing kill switches. But practically, such a system has a major limitation that the organism could proliferate in any environment upon excretion (i.e. the default condition is permissive because aTc would be absent). That makes the 2nd input of temperature-response a very critical component in ensuring that genetically modified microbes are not released in the wild after excretion. The authors should further discuss the rationale for their design and inputs more in the introduction (pros and cons), both in reference to traditional kill switches and in motivating why they looked at the 2nd input, because it currently seems like an afterthought.

We thank the reviewer for their insightful comment. We have correspondingly reorganized and edited the introduction to emphasize the reviewer's distinction regarding traditional biocontainment circuit designs that rely on an element inherent or supplied to the permissive environment, which when absent then allows the expression of the lethal kill switch components, and contrast this to our 'inverse' kill switch design in which an externally supplied signal induces expression of the lethal components. We further clarify in the introduction that our final deliverable is indeed the 2-input killswitch that includes a temperature-sensitive response, but that to additionally control proliferation of the engineered probiotic within the gut, we chose such an 'inverse' kill switch design in which the restrictive condition inside the gut is externally supplied. Finally, we clarify in the introduction that we took a step-wise design approach such that the single-input kill switch circuit is the foundation for the 2-input kill switch, in that the latter replaces constitutive expression of TetR with temperature-dependent expression. Thus, while much of the iterative optimization for stability and efficacy was performed on the single-input circuit, these optimization efforts directly benefited our final deliverable of the 2-input kill switch (LL74-117).

2. The literature survey of biocontainment systems is good but misses a few more recent references, particularly in regard to where stability was investigated. The Silver group has reported more recent kill switches (Stirling et al 2020). Similarly, the Church group has reported stability evaluation of auxotrophy based on synthetic amino acids (Kunjapur et al 2021). Also, the introductory text suggests that all auxotrophy can be overcome by cross-feeding but this limitation does not apply when synthetic amino acids are used.

These are indeed highly relevant publications and we apologize for the omission. We now include cited references to this prior work in our introduction (LL74-117, References 18 and 21). In addition, we have edited the introduction to specify that auxotrophy can be overcome by cross-feeding only in the case where the essential compound or viable alternatives are natively present (LL88-90).

3. There were a few figures that were difficult to understand with the level of explanation currently provided. Figure S1D was one such example. Please elaborate there.

We thank the reviewer for pointing out the lack of detail in figure legends. We edited for clarity and added more in-depth descriptions to the legends for Figures 1F, 2A-C, 3G, 5C S1C-D, S2, S3A-C, S3F and S4E.

4. More information should be provided in relation to the antibiotic-independent *infA* construct. It is currently brought up in Line 204, with ambiguity about the cloning site and strength of the constitutive promoter (it is possible that I missed this). Additionally, it looks like antibiotic was still added in some experiments that used this construct (Figs S2B, S2C, S3C), so does the construct still contain an antibiotic resistance marker?

We apologize that this information regarding construct design was not clearly provided. *InfA* was incorporated onto the gRNA plasmid between the *tetR* gene and *rrs-2* gRNA cassette using Gibson Assembly. We have deposited sequence files for all generated plasmids (Table S5) to Mendeley Data where they will be made available prior to publication and can be accessed to acquire more information. *infA* was also incorporated onto the plasmid with an intermediate strength constitutive Anderson promoter. This information was added to the results (LL211) and the sequence of the promoter (p07) was previously included in Table S6. Regarding the antibiotic resistance marker, yes, the *infA*-expression gRNA plasmids still harbored an antibiotic resistance marker. We left the marker on the plasmid to allow for colony forming unit quantification from mixed-strain samples (e.g., fecal samples). This was clarified in the results (LL213-214). In future work we will remove the antibiotic resistance marker, which we now comment on in the discussion (LL530-532).

5. aTc concentrations vary in Fig 1C/1H, which is alluded to in the Fig 1 caption. Could the authors comment on the reason for this change – was it simple assay optimization or something else?

Thank you for commenting on this change. Yes, we increased the aTc concentration between figures 1C and 1H due to assay optimization. More specifically, we found improved killing efficiency at higher aTc concentrations when we completed the transfer curve for the *ile-2* kill switch (Figure S1E), so we increased the concentration in subsequent experiments. We have clarified this in the legend for Figure 1H (LL631-632).

6. The specific gRNA targets that are used in various assays could use more clarity. In Fig 1C, *groL-2*, *ile-2*, and *rrs-2* all seem comparable as targets. It is clear from the text and from the figure that *groL-2* and *ile-2* are pursued further as targets first (e.g. Fig 1H). Then, the text seems to make it clear that *rrs-2* is tested to improve the *infA* construct; but neither Fig 2 or S2 seem to specify the gRNA targets (they say X3 instead of X2, but in all other places this refers to redundancy, whereas these are distinct targets). After Fig 2, it is unclear if all 3 gRNA targets are consistently used in the subsequent kill switches. It is also unclear what the Mut14 variant is (what has been mutated, to what sequence?).

Thank you for bringing these points to our attention. Since the three different gRNAs are not identical, we agree that it is not accurate to group them as X2 or X3. As such, we have changed the circuit schematics and figure captions in Figures 2A-C, S2, and S3A to specify the gRNA targets used. After Figure 2, all subsequent kill switches maintain the same three distinct gRNAs. This is made clear in Table 1. However, we would like to clarify that our use of the term redundancy refers distinctly to functional redundancy (guiding Cas9 to the genome), rather than redundant sequences. The Mut14 variant was obtained

by adding the *rrs-2* gRNA to the optimized *infA* construct where the Ptet promoter was incorporated with a randomized -35 site. Since the *rrs-2* expression cassette was inserted onto the plasmid as a library, there was not an initial construct that was then mutated. However, we have included the sequence of the -35 site for the Ptet promoter in Mut14 as Figure S3G.

7. This is not a request for the authors to perform additional experiments, however this reviewer is very curious whether other ratios of co-delivered strains were tested besides 1:1, even in the *in vitro* assay? Given how effective and important this approach is, it would be nice to know how tuning of the ratio influences escape.

The reviewer brings up a great point of discussion. We had previously obtained data during *in vitro* assay optimization that suggested a correlation between control:kill switch ratio and kill switch efficiency. However, we had not tested the different ratios using a kill switch with the Δ rpdu knockouts. Out of interest, we performed the competitive *in vitro* assay using a variety of control:CRISPRs Δ rpdu ratios and found that the co-culture method provided robust protection against kill switch inactivation and escape. We found that the kill switch strain was completely eliminated from co-cultures using control:CRISPRs Δ rpdu ratios as low as 1:1000. As such, the co-delivery technique may provide robust protection against stochastic variations in sample preparation and differing colonization efficiencies between the two strains. Furthermore, the ratio used in clinical practice can likely be lower than the 1:1 tested *in vivo* here to increase the amount of the engineered strain delivered and maximize the therapeutic potential. However, we can't guarantee that these different ratios will perform similarly *in vivo*. We have included these findings as Figure S4C and discussed them in the results (LL279-284).

8. In Fig 3G, the 24 hour (del)rpdu error bar is massive, and at other time points the performance is equivalent to the other strain. Can any meaningful conclusions be drawn from this data?

We inadvertently left out the statistical analysis from this figure; thank you for pointing out this omission. We performed a mixed model ANOVA with Sidak's multiple comparisons and showed that at the 24hr timepoint, the percent non-functional values for the CRISPRs and CRISPRs Δ rpdu strains are significantly different (p-value of 0.0042). We have now included the results of this test in Figure 3G.

9. In Fig 3J, why is there a lack of CFUs/mg in the control settings? Is this an indication of a lack of colonization by a codelivered strain? This seemed unexpected but it is possible that I missed something.

While there was indeed detectable CFU/mg in cecal contents for the control strain in both aTc and control water arms (for 3 out of 4 replicate mice in each case), we agree that the CFUs/mg were significantly lower than in other experiments. To improve visibility we have updated the y-axis of panel 3J to a log scale, which is more consistent with the other representations of cecal data in the manuscript so we thank the reviewer for alerting us to this discrepancy. While it is briefly mentioned in the results (LL335-336), we have also edited the Figure 3 legend to clarify that cecal contents were sampled on day 8 after gavage, not at 48 hrs which may have been erroneously inferred from the layout of Figure

3. However, day 8 cecal titers overall were indeed low (maximum of 60 CFU/mg in any single sample) in this experiment compared to the experiment with the 2-input strain (Figure 5 and Figure S7); this discrepancy is perhaps attributable to variation in the gut microbiomes of commercial mice, which can vary based on litter and housing location, and correspondingly may provide different levels of colonization resistance despite antibiotic treatment. We clarify in the results, however, that cecal titers were not significantly different between the control strain in either treatment arm and the CRISPRks Δ rpdu strain in the control water arm (LL340-341). In addition, we clarify in the results that in one technical replicate of the CRISPRks Δ rpdu + aTc arm, plating of undiluted fecal homogenate did result in detectable growth of one colony (LL334-336).

10. 5B: The OR behavior (atC -, 2-input CRISPRks) is visibly weaker than the AND gate expected upon excretion, which is fair. However, is this good enough for biocontainment of an engineered probiotic? It seems like it will proliferate upon excretion until aTc is added.

Indeed, in the absence of aTc exposure within the gut there was an increasing incidence over time of biocontainment failure for the 2-input (temperature-responsive) CRISPRks, which was particularly apparent at the end of the experiment on day 7 (168 hr) and day 8 (sacrifice). It is evident from sequencing of isolates recovered from other arms in this experiment that mutations in the *t1pA* cassette were not uncommon (Figure 5D), though we unfortunately are not able to attribute the biocontainment failure in this arm to specific mutations. We note that even *in vitro* the 2-input kill switch did not achieve full efficacy when induced with temperature alone (Figure 4B), in contrast to the single-input kill switch when induced with aTc (Figure 2C). This suggests that P*t1pA*-driven expression of *tetR* is leaky even at low temperatures or that the TetR degradation rate is not sufficiently high such that the 2-input CRISPRks strain requires both induction modalities to perform robustly. As such, a limitation of our 2-input kill switch design is that the two inputs are not orthogonal, and not synergistic compared to the single-input design since they feed into the same kill switch mechanism. We now comment on these limitations in the discussion (LL545-567). However, with the co-delivery approach, the control strain, which is excreted from the mice simultaneously with the kill switch strain, may also provide a degree of protection against unwanted proliferation of 2-input CRISPRks Δ rpdu microbes with inactivated kill switches.

11. 5F: As mentioned in the text, the 2-input kill switch leads to escape under one of the conditions measured – the 72 h timepoint. The discussion of this result seems limited. What are the implications for how this switch could be applied, and what next steps if any might be needed for improvement? Since intra-niche competition is already being used, could the authors offer a brief recommendation on what else should be considered by future researchers to prevent escape?

The reviewer is correct--even with both induction modalities and intra-niche competition, 24 hr growth assays inoculated from the 72 hr timepoint samples resulted in a mean of 583 CFU/mL of the 2-input CRISPRks. Using control strain titers from the same mice at the same timepoint, we estimate a room temperature doubling time of 115 minutes, which would imply a baseline CRISPRks titer of 0.1 CFU/mL, or ~100 CFU in the entire fecal sample given our inoculation scheme. This was not captured when plating directly from feces, in which approximately 2% of the entire fecal sample is plated at the 1X dilution. This highlights the importance of survival assay sensitivity in the development of kill switches. Escape events such as these could be reduced by addition of a truly orthogonal

kill switch mechanism, and optimization of the *tlpA* cassette to reduce leakiness at room temperature, or of TetR degradation rates, as discussed in the response to comment #10 above. We have added these comments and future directions to the discussion section (LL545-567).

Reviewer #2 (Remarks to the Author):

In this manuscript, the authors reported two “CRISPR-based kill switches” in *E. coli* Nissle 1917 (EcN), that could selectively terminate the host cells, upon anhydrotetracycline induction or exposure to low temperature (<33oC). To ensure effective killing by the kill switches, the authors performed extensive optimization on the genetic circuits and the chassis strain, including development of antibiotic-independent plasmid system, implementation of functional redundancy with the Cas9 expression cassette, modulation of bacterial SOS response and provision of intra-niche competition by a closely related strain. The optimized strains with embedded kill switches were then tested in the mice model to show efficient elimination of EcN, upon induction. Overall, the manuscript is well-written. The findings of the study are well-presented, and the conclusions drawn by the authors are well-founded. Please find below my comments.

We thank the reviewer for this very positive comment.

1. Would the implementation of kill switches potentially impair the capability of therapeutics production in EcN? The authors have edited EcN genome to a great extent to ensure the efficiency of the kill switches, which included integration of the Cas9 expression cassette, deletion of essential gene *infA* and deletion of SOS response genes. Considering that these modifications can affect essential cellular functions, have the authors considered the impact of these modifications on the therapeutic production capability of EcN?

The reviewer brings up a great potential concern of the kill switch, in that the therapeutic potential of the kill switch strain needs to be considered. Therapeutic proteins are often expressed both chromosomally and on plasmids. As such, kill switch strains should be able to express heterologous proteins at comparable levels to wild-type EcN. To compare the production capabilities of the kill switch strains, we compared expression levels of constitutively expressed GFP in wild-type EcN, the no gRNA control, the aTc-responsive CRISPRks, and the aTc- and temperature-responsive 2-input CRISPRks strains using both a genome-integrated cassette and a plasmid-based cassette for GFP expression. The kill switch strains demonstrated GFP expression levels comparable to wild-type EcN with both expression methods. The results were added as Figure S8 and explained in the results section (LL460-469).

2. As the kill switches are intended to serve as a biocontainment mechanism for engineered therapeutic EcN strains, I suggest the authors to characterize the impact of the kill switches on protein productions in EcN. This can be done by either comparing the expression level of native proteins or co-expression of reporter proteins in the engineered EcN strain.

As discussed in the above comment, we agree with the reviewer’s suggestion to consider the impact of the kill switches on protein production in EcN and accordingly characterized their protein production potential using constitutive GFP expression as a proxy for

therapeutic proteins (Figure S8). Both kill switch strains achieved GFP expression levels equivalent to wild-type EcN using both a genome-integrated expression cassette and a plasmid-based expression cassette, suggesting that the extensive engineering and biocontainment systems do not adversely affect the ability of the cell to produce recombinant proteins.

3. One of the major concerns in the applications of genetically modified microorganisms is a potential horizontal gene transfer that can occur amongst engineered strains to surrounding native microbiome. Such a phenomenon has been previously reported in two separate studies (<https://pubmed.ncbi.nlm.nih.gov/17014680/>; <https://pubmed.ncbi.nlm.nih.gov/18238887/>). In the case of kill switches, would the genetic materials, especially plasmids, be released to the native microbiome following bacterial cell death? If so, will it increase the chances of horizontal gene transfer? Authors should discuss such possibilities and elaborate on how to minimize horizontal gene transfer event that can occur with these kill switches.

The reviewer makes a great point that biocontainment of genetically engineered organisms must not neglect biocontainment of the corresponding recombinant DNA. Our kill switch design addresses the risk of horizontal gene transfer of plasmid-borne antibiotic resistance genes in that the only plasmid-borne elements are the Tet repressor, which alone does not confer resistance, and the gRNAs, whose transcription should have no effect without the Cas9 cassette, and which carry inherent specificity to the EcN genome. We maintain the plasmid through use of an *infA*-knockout strain, where *infA* is complemented on the plasmid, precluding the need for antibiotic-driven plasmid maintenance. We do include SpecR and ChIR on plasmid constructs in order to differentiate the CRISPRs and control strains, respectively, and to quantitate EcN from stool. This inclusion was a research tool, and any clinical iteration of these probiotic strains will omit these resistance genes. We have included these discussion points in the text, including cited references to the two publications the reviewer kindly provided (LL522-532).

4. The authors have engineered a number of EcN strains with various genetic circuits. To make it easier for the readers to follow, the authors can include the corresponding strain names in the schematics of the genetic circuits in the figures (eg. in Figure 1A, Figure 2A B C, and Figure 4A).

Thank you for this great suggestion. We agree that it was difficult to follow the different strains and genetic circuits through the various iterations of the kill switch. To address this, we have added kill switch names to the suggested figures which should help readers track which kill switch strain is relevant to each section. Readers can also further refer to Table 1 for additional details on the different strains and genetic circuits used in each figure.

5. Line 257-259 “Thus, we hypothesized that a nutrient- and growth-limited environment impairs the DNA-cleavage rate of the kill switch and allows the survival of daughter cells with an induced SOS response and elevated kill switch inactivation rate.” Further explanation is needed to understand this hypothesis. Please explain why the growth-limited environment impairs the DNA-cleavage rate and how this is related to SOS response genes. Secondly, if the DNA-cleavage rate was indeed impaired, shouldn't it lead to a weaker or slower SOS response?

Thank you for bringing up this important discussion. *E. coli* has been shown to have lower per cell protein expression in growth-limited environments than in rich medium {PMID: 26641532}. We now included this citation in the results (LL266-267). We have also included new data (Figure S8) confirming that the kill switch strains have lower single cell expression of GFP in growth-limiting medium than in rich medium when the gene is genome integrated (as Cas9 is here). As such, it is likely that expression of the kill switch is reduced in growth-limited medium, causing reduced DNA-cleavage rates. However, it has also been previously shown that even a single DNA double strand break strongly induces the SOS response in *E. coli* {PMIDs: 15948952; 17529976}. As such, the lower kill switch expression wouldn't necessarily weaken the activity of the SOS response. We also included these citations in the results (LL263). We have included these arguments in both the results (LL262-269) and discussion (LL495-500) sections.

6. Line 364 -366, "We tested over 500 2-input CRISPRs Δ rpdu variants". A few different methods were applied to improve the temperature response in the engineered EcN strain. Please provide more detailed and descriptive term than "CRISPRs Δ rpdu variants" to provide better understanding of the range of EcN strain variants being tested?

Thank you for pointing out this lack of clarity. The "CRISPRs Δ rpdu variants" refer to strains in the combined ribosome binding site and degradation tag library described in the prior sentence (LL379-381) and Figure 4C. We added text to the cited results sentence to better explain what the different variants are (LL381-383).

7. Line 802 - 804, streptomycin was administered to mice to enable EcN colonization. The reference cited, however, used the same treatment to enable the colonization of a *Salmonella enterica*. Please provide more relevant citation (i.e. for EcN colonization).

Thank you for pointing it out. Indeed, streptomycin treatment is often used in colonization models for *Salmonella enterica*, but has also been extensively used in colonization models for pathogenic and commensal strains of *E. coli* {PMIDs: 8359904; 2196227; 19364832; 18180286; 23349773; 29358683}. Our group has previously used streptomycin treatment of mice to enable the colonization of EcN {PMID: 30926240}. Streptomycin treatment helps overcome colonization resistance observed in conventional mouse models by depleting commensal facultative anaerobes. Additionally, we report optimization of our streptomycin-treatment colonization model, and compare it to no antibiotic treatment as well as a carbenicillin treatment recently reported as an alternative colonization model for EcN {PMID: 33301298} (**Figure S5**), ultimately choosing 20 mg streptomycin followed by a 72h rest period prior to EcN gavage as the optimal method to ensure equal colonization dynamics of our control and CRISPRs strains. We have added these *E. coli*-relevant references to the methods section (LL845-847).

Reviewers' Comments:

Reviewer #1:

Remarks to the Author:

The revised manuscript addresses all my concerns. The authors have improved the clarity for the motivation of their system, the experimental constructs, and the experimental conditions.

Reviewer #2:

Remarks to the Author:

The authors have adequately addressed my comments.